Review

# Cholesterol and COVID-19—therapeutic opportunities at the host/virus interface during cell entry

Thomas Grewal[1] , Mai Khanh Linh Nguyen[1], Christa Buechler[2]

The rapid development of vaccines to combat severe acute respiratory syndrome coronavirus 2 (SARS-CoV-2) infections has been critical to reduce the severity of COVID-19. However, the continuous emergence of new SARS-CoV-2 subtypes highlights the need to develop additional approaches that oppose viral infections. Targeting host factors that support virus entry, replication, and propagation provide opportunities to lower SARS-CoV-2 infection rates and improve COVID-19 outcome. This includes cellular cholesterol, which is critical for viral spike proteins to capture the host machinery for SARS-CoV-2 cell entry. Once endocytosed, exit of SARS-CoV-2 from the late endosomal/lysosomal compartment occurs in a cholesterol-sensitive manner. In addition, effective release of new viral particles also requires cholesterol. Hence, cholesterol-lowering statins, proprotein convertase subtilisin/kexin type 9 antibodies, and ezetimibe have revealed potential to protect against COVID-19. In addition, pharmacological inhibition of cholesterol exiting late endosomes/lysosomes identified drug candidates, including antifungals, to block SARS-CoV-2 infection. This review describes the multiple roles of cholesterol at the cell surface and endolysosomes for SARS-CoV-2 entry and the potential of drugs targeting cholesterol homeostasis to reduce SARS-CoV-2 infectivity and COVID-19 disease severity.

## Introduction

COVID-19 caused by infections with SARS-CoV-2 are accompanied by heterogeneous indications ranging from asymptomatic infections to serious and life-threatening manifestations, representing a major challenge to foretell COVID-19 disease outcome (Bean et al, 2023). Besides advanced age and male sex (Booth et al, 2021; Sieurin et al, 2022), other predictors for adverse outcomes in SARS-CoV-2–infected patients include obesity, cardiovascular disease, hypertension, type 2 diabetes, and other diseases such as cancer, kidney diseases, obstructive pulmonary disease, or pre-existing cerebrovascular and respiratory diseases (Booth et al, 2021; Cheng et al, 2021; Ng et al, 2021; Westheim et al, 2021).

### Targeting SARS-CoV-2 entry and trafficking along the endocytic pathway

Although the rapid development of vaccines greatly reduced the severity of COVID-19, the continuous emergence of new virus variants underscores antivirals to remain essential for the management of SARS-CoV-2 infections. Antivirals that target viral proteins include the clinically approved drug nirmatrelvir, which inhibits the viral major protease ($M^{pro}$), and consequently, SARS-CoV-2 replication. Paxlovid combines nirmatrelvir with the HIV protease inhibitor ritonavir. The latter inhibits the cytochrome P450 3A4 enzyme, which increases the half-life of nirmatrelvir, providing antiviral activity against existing coronavirus variants (Blair, 2023).

Lopinavir is another potent HIV-1 protease inhibitor, leading to the production of immature, non-infectious virions. Although its combination with ritonavir increased bioavailability, this drug combination was associated with worse clinical outcomes in hospitalized COVID-19 patients (Babayigit et al, 2022).

Alternatively, the repurposing of drugs that target host factors hijacked by the virus shows promise. This includes the antiviral and immunomodulatory activities of the Food and Drug Administration (FDA)–approved antiparasitic ivermectin. Multiple drug–protein interactions with viral and host proteins at the cell surface and cytoplasm interfere with virus entry and propagation and provide anti-inflammatory activities targeting JAK/signal transducer and activator of transcription (STAT), phosphatidylinositol-3 kinase/protein kinase B (PI3K/Akt), and nuclear factor kappa-light-chain enhancer of activated B cells (NF-kB) signaling pathways. However, evidence of its therapeutic benefits to treat COVID-19 is still lacking (Shafiee et al, 2023).

Angiotensin-converting enzyme 2 (ACE2) is the major host cell receptor enabling SARS-CoV-2 to attach to the host cell surface. In the respiratory system, ACE2 was not detectable or expressed at low levels in some cells but was present in many other cells and tissues such as enterocytes, gallbladder, cardiomyocytes, ductal cells, and vasculature, indicating that high ACE2 expression is not an indicator

---

[1]School of Pharmacy, Faculty of Medicine and Health, University of Sydney, Sydney, Australia    [2]Department of Internal Medicine I, Regensburg University Hospital, Regensburg, Germany

Correspondence: christa.buechler@klinik.uni-regensburg.de

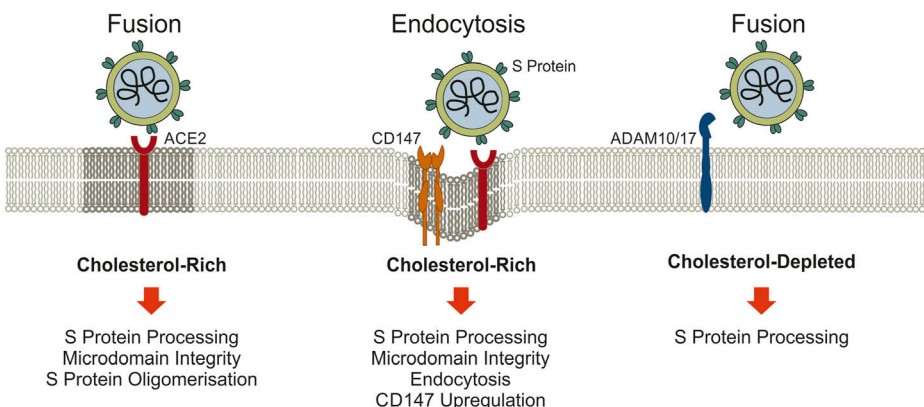

**Figure 1. Cholesterol at the cell surface influences SARS-CoV-2 cell entry.**
(i) Cholesterol depletion displaces ACE2 from rafts and reduces SARS-CoV-2 docking and ACE2-dependent viral envelope fusion with the host membrane as well as endocytotic virus uptake. Spike protein oligomerization improves viral membrane fusion and is enhanced by cholesterol. (ii) Cholesterol loading of cells induces CD147 levels, which can mediate SARS-CoV-2 endocytosis and contribute to the higher risk for severe COVID-19 in dyslipidemia. (iii) ADAM10/17 contribute to S protein processing and are activated in cholesterol-depleted cells, a route that may overcome reduced infection efficacy when cell surface cholesterol levels are low. Abbreviations: ADAM 10/17, A Disintegrin And Metalloproteinase 10/17; ACE2, angiotensin-converting enzyme 2; CD147, cluster of differentiation 147; SARS-CoV-2, severe acute respiratory syndrome coronavirus 2; S protein, spike protein.

of increased susceptibility to infection (Hikmet et al, 2020). Consistent with these findings, only a relatively small number of ACE2-positive cells were found in the human respiratory tract. Further indicating that ACE2 protein levels per se do not determine susceptibility to COVID-19 infection, increased ACE2 protein levels were found in individuals at lower risk of severe COVID-19, such as children and healthy controls (Ortiz et al, 2020).

Other host cell receptors for SARS-CoV-2 include neuropilin receptors, C-lectin–type receptors, cluster of differentiation 147 (CD147), heparan sulfate proteoglycans, dendritic cell-specific intercellular adhesion molecule-3-grabbing non-integrin (CD-SIGN), liver/lymph node-specific intercellular adhesion molecule-3-grabbing integrin (L-SIGN), macrophage galactose-type lectin, glucose-regulated protein 78, AXL receptor tyrosine kinase, and T-cell immunoglobulin and mucin domain protein 1 (reviewed in Cesar-Silva et al [2022]). The virus then enters the cell via fusion with the plasma membrane or clathrin-dependent and -independent endocytosis (Al-Horani et al, 2020; Sumbria et al, 2020). Current vaccines targeting the initial interaction of viral and host cell proteins at the cell surface have greatly contributed to combat COVID-19 disease.

Internalized viruses trafficking along the endocytic pathway reach late endosomes/lysosomes (LE/Lys) and then exit this compartment to replicate and build new viral particles in the cytoplasm. Hence, more than 50 proteins and adaptors of the endocytic machinery showed therapeutic potential, with gene knockdown approaches of key players blocking SARS-CoV-2 entry (dynamin, clathrin, and caveolin) or small molecules such as dynasore (dynamin), pitstop 2 (clathrin-coated pit formation), chlorpromazine (clathrin-dependent endocytosis), rottlerin, amiloride (macropinocytosis), and LY294002 (phagocytosis) reducing SARS-CoV-2 infection (Bayati et al, 2021; Li et al, 2021b; Cesar-Silva et al, 2022; Alkafaas et al, 2023). Meplazumab, an antibody against the SARS-CoV-2 receptor CD147 (Fig 1), inhibited virus entry and amplification in fibroblasts, reducing the production of extracellular matrix proteins contributing to pulmonary fibrosis (Wang et al, 2020a; Wu et al, 2022). Dominant-negative mutants of Rab5 and Rab7 GTPases, which regulate the trafficking of vesicles along the endocytic pathway, also displayed an ability to block SARS-CoV-2 infection (Zang et al, 2020).

In addition, the targeting of cathepsins, endolysosomal cysteine proteases that are required for the processing of internalized virus to enter the cytoplasm, can lower SARS-CoV-2 infection. This includes cysteine protease inhibitors such as E64d and K11777 and cathepsin L–specific compounds Z-FY9t-Bu-DMK and teicoplanin (Huang et al, 2006; Zhou et al, 2015, 2016).

Furthermore, lysosomotropic agents, which cause lysosomal membrane permeabilization and dysfunction, have been considered to inhibit coronavirus infection This comprises hydroxychloroquine, which is used to treat malaria and several autoimmune diseases, and has antiviral effects due to its ability to compromise endo- and lysosomal acidification. Although large-scale randomized controlled trials failed to show any survival benefit of hydroxychloroquine for COVID-19 patients, it was approved by the FDA for emergency COVID-19 therapy (Self et al, 2020). Of note, in several clinical studies with >500 participants, twice or more the FDA-recommended doses of hydroxychloroquine were used, yet these studies did not provide conclusive evidence of the effectiveness of hydroxychloroquine in treating COVID-19 (Axfors et al, 2021; Kumar et al, 2021). Yet, hydroxychloroquine, as well as chloroquine, can severely compromise vital LE/Lys functions, and very high doses can lead to cell death and strong side effects and toxicity profiles.

Nevertheless, several acidification inhibitors block the exit of endocytosed SARS-CoV-2 from LE/Lys to significantly reduce infectivity of SARS-CoV-2 and other coronaviruses (Hoffmann et al, 2020; Prabhakara et al, 2021). However, the suitability of niclosamide, bafilomycin A1, and NH$_4$Cl to treat SARS-CoV-2 infections requires further fine-tuning, as de-acidification results in profound changes and dysfunction of the LE/Lys compartment, many of those mandatory for fundamental cellular functions. Alternatively, inhibiting the maturation of LE to Lys, using apilimod and YM201636, effectively reduced SARS-CoV-2 infections (Kang et al, 2020). Hence, the host's cell surface receptors and endocytic compartment provide a variety of therapeutic avenues to lower SARS-CoV-2 infection.

## Role of cholesterol at the plasma membrane for SARS-CoV-2 infection

Various human pathogenic viruses depend on the hosts' lipid metabolism for infection and replication. Patients with severe

infectious diseases have low levels of low-density lipoprotein (LDL) and high-density lipoprotein (HDL), and cholesterol loading of immune cells and most likely further cells and tissues seems to contribute to hypolipidemia (Wang et al, 2023). Accordingly, serum lipoprotein levels have prognostic value to predict the severity and mortality of patients with COVID-19 (Liu et al, 2021).

Cellular uptake of LDL- and HDL-derived cholesterol rapidly influences cholesterol levels at the plasma membrane and endocytic vesicles including LE/Lys, and dyslipidemia has been implicated to support SARS-CoV-2 infection by altering membrane lipid composition and interfering with immune cell function (Makhoul et al, 2022). Importantly, cell surface binding, cellular uptake, and propagation of SARS-CoV-2 are intimately linked to cellular cholesterol metabolism and distribution, providing therapeutic opportunities (Wang et al, 2022; Talukder et al, 2023).

The plasma membrane contains 60–80% of total cellular cholesterol, which represents 30–40% of all plasma membrane lipids. Moreover, cell surface cholesterol preferentially partitions in cholesterol-rich membrane domains (lipid rafts) containing receptors and functioning as signaling hubs to transmit environmental cues (Mesmin & Maxfield, 2009; Enrich et al, 2015). These specialized cholesterol-rich microdomains are heterogenous in terms of their lipid and protein content, often conferring different functions. For instance, lipid rafts enriched in either saturated ceramide-containing glycolipids such as monosialotetrahexosylganglioside 1 (GM1) or phosphatidyl-inositol 4,5 bisphosphate (PIP2) containing lipid clusters exist and movement of GM1-localized membrane proteins to PIP2 lipid clusters upon localized changes in membrane cholesterol levels may affect their biologic function (Yuan & Hansen, 2023). Lipid rafts also contain receptors that are recognized by viruses for cell entry, and disruption of lipid rafts using cholesterol-depleting agents such as methyl-$\beta$-cyclodextrin (M$\beta$CD), which can remove up to 80% of cellular cholesterol, strongly reduced the entry, infectivity, and release of murine coronavirus, SARS-CoV, and SARS-CoV-2 (Thorp & Gallagher, 2004; Li et al, 2007; Li et al, 2021b) (reviewed in Fecchi et al [2020]). ACE2 locates to lipid rafts and statin-induced cholesterol depletion reduced SARS-CoV-2 internalization in human epithelial lung adenocarcinoma Calu-3a cells (Cesar-Silva et al, 2022) (Fig 1). As ACE2 protein levels remained unchanged upon M$\beta$CD treatment, the association of ACE2 with cholesterol-rich membrane domains appears important for viral entry (Lu et al, 2008) (Fig 1). Interestingly, in a diabetic mouse model characterized by elevated cholesterol with increased age and disease, ACE2 was predominantly found in cholesterol-rich GM1 membrane structures that facilitate endocytosis and cholesterol depletion caused ACE2 to move from GM1 lipid to PIP2 lipid domains, thereby decreasing virus uptake and infectivity (Wang et al, 2023). Super-resolution microscopy also revealed a relationship between the spike protein and PIP2 clusters (Raut et al, 2022). In addition, hydroxychloroquine altered the association of ACE2 with GM1- and PIP2-containing microdomain clusters in cells with high and low cholesterol, respectively (Yuan et al, 2022). Also, avasimibe-mediated inhibition of acyl-CoA:cholesterol acyltransferase (ACAT)–mediated cholesterol esterification disrupted ACE2 association with GM1 lipid rafts and inhibited SARS-CoV-2 pseudoparticle infection (Wing et al, 2023).

Besides using the cholesterol-depleting agent M$\beta$CD, apolipoprotein E (ApoE) is an endogenous cholesterol transport protein that in the non-lipidated form removes cholesterol, whereas the lipidated form is a physiological cholesterol donor for cells. Most relevant for COVID-19, lipidated ApoE induced ACE2 movement to GM1 rafts and enhanced virus infection (Wang et al, 2023). Three apoE isoforms exist in humans—apoE2, apoE3, and apoE4—and it is in this order that these genotypes are associated with increasing LDL-cholesterol levels and coronary artery disease (Tudorache et al, 2017). Homozygous apoE4 carriers had an increased risk of severe COVID-19 infection, independent of preexisting comorbidities (Kuo et al, 2020). As the apoE4 genotype is associated with higher tissue cholesterol levels, oxidative stress, and inflammation, this may all contribute to greater susceptibility to SARS-CoV-2 infection and COVID-19 severity (Gkouskou et al, 2021).

Cholesterol-depleting cyclodextrins reduced binding of the S1 subunit of the viral spike (S) protein to ACE2 in Wuhan-Hu-1, Delta, and Omicron virus variants (Kovacs et al, 2023), which was reinforced by cholesterol depletion studies in SARS-CoV–infected Vero-E6 (African green monkey epithelial kidney cell line) and Calu-3a cells (Li et al, 2007; Lu et al, 2008; Wang et al, 2020b). These observations might relate to clinical settings, as pantethine, a vitamin B5 derivative, which reduces total and LDL-cholesterol, was associated with reduced SARS-CoV-2 infection (Abou-Hamdan et al, 2023).

Besides raft-dependent endocytosis, cholesterol can also support fusion of the viral envelope with raft domains at the plasma membrane (Ripa et al, 2021; Sanders et al, 2021; Cesar-Silva et al, 2022) and several cholesterol-sensitive players appear to be involved in the fusion process (Tang et al, 2021a). After docking of the S protein to ACE2, fusion of the host and viral membranes facilitate viral entry. This route depends on the processing of the viral S2 subunit by transmembrane serine protease type II (TMPRSS2) (Bestle et al, 2020).

Alternatively, fusion of endocytosed viruses with endosomal and cholesterol-rich membranes requires cathepsin L to cleave the S protein (Huang et al, 2006; Li et al, 2007; Zhou et al, 2015, 2016; Bestle et al, 2020; Yang & Shen, 2020). Besides furin and TMPRSS2 (Al-Horani et al, 2020; Nejat et al, 2023), a disintegrin and metalloproteinase (ADAM) 10 and 17 contribute to the processing of the viral S2 subunit (Jocher et al, 2022). Showing potential as therapeutic targets, ADAM protease inhibitors lowered SARS-CoV-2 infection in lung cells (Jocher et al, 2022) and the FDA-approved serine protease inhibitor clamostat blocked SARS-CoV-2 entry by inhibiting ACE2 and TMPRSS2 (Hoffmann et al, 2020). Platycodin D from *Platycodon grandifloras* redistributed membrane cholesterol, thereby antagonizing fusion of viral and cell membranes and preventing SARS-CoV-2 entry via endosomal and TMPRSS2 pathways (Kim et al, 2021b). Interestingly, cholesterol depletion enhanced ADAM10- and 17-mediated substrate cleavage (Reiss & Bhakdi, 2017), yet if this impacts on SARS-CoV-2 infection remains to be clarified (Fig 1).

Furthermore, membrane cholesterol accelerates the formation of oligomeric S protein structures that enhance S2 activity (Meher et al, 2023) (Fig 1). Host site-1 protease (S1P)–mediated activation of sterol regulatory element–binding protein 2 (SREBP2), the major transcription factor that controls cholesterol homeostatsis,

increased SARS-CoV-2 fusion with the cell membrane due to en-hanced S2 processing in a HeLa (human cervical carcinoma) cell model (Essalmani et al, 2023).

Of note, MβCD-sensitive entry mechanism mediated by the spike protein, yet independent of clathrin-coated pits and caveolae (Wang et al, 2008), further illustrate the diverse roles how cellular cholesterol at the cell surface can modulate SARS-CoV-2 cell entry (Wang et al, 2008; Essalmani et al, 2023).

Besides cholesterol-depleting agents, oxysterols such as 25-hydroxycholesterol can also reduce the availability of cell surface cholesterol, with inhibitory consequences for viral infectivity (reviewed in Mao et al [2022]). In Calu-3a cells, Caco-2 cells, and lung organoids, 25-hydroxycholesterol interfered with the fusion of SARS-CoV-2 with the cell surface (Wang et al, 2020b), most likely because of the transfer of cell surface cholesterol to lipid droplets. Alternatively, in HEK293 (human embryonic kidney) cells, 25-hydroxycholesterol accumulated in LE/Lys and blocked cholesterol export from this compartment, which appeared to inhibit SARS-CoV-2 spike protein-mediated fusion with the LE/Lys membrane (Zang et al, 2020). Notably, cholesterol-25-hydroxylase, which generates 25-hydroxycholesterol, is stimulated by interferon, up-regulated by SARS-CoV-2 infection and in COVID-19 patients, in-dicating that the generation of this oxysterol is part of the innate immune response to combat SARS-CoV-2 propagation (Mao et al, 2022).

## Cholesterol accumulation in LE/Lys and the role of Niemann–Pick type C1 protein for infection

As outlined above, disruption of cholesterol-sensitive mechanisms at the cell surface can interfere with SARS-CoV-2 cell entry (see also Fig 1). Yet, inhibition of protease TMPRSS2 at the plasma membrane in combination with blocked endolysosomal cathepsins completely abrogated SARS-CoV-2 entry (Hoffmann et al, 2020), identifying the endocytic pathway as another prominent route for SARS-CoV-2 cell entry (Ou et al, 2020; Tang et al, 2020), and a plethora of drugs that interfere with endocytic trafficking of SARS-CoV-2 have been de-scribed in the previous chapter above. Remarkably, more recent variants of SARS-CoV-2, such as the Omicron BA.1 variant, displayed a lower efficiency in using the cellular protease TMPRSS2 and relied more on cell entry through endocytic pathways that are sensitive to cholesterol (Meng et al, 2022). A recent study highlighting the re-quirement of an acidic endosomal environment for early variants of SARS-CoV-2 also point at the potential to target late endosomal cholesterol (Kreutzberger et al, 2022). Hence, drugs targeting endocytic pathways and late endosomal cholesterol may be ef-fective for therapy of patients infected with both recent and earlier variants.

In fact, many zoonotically transmitted viruses, in particular enveloped viruses including SARS-CoV-2 (Tang et al, 2020), hijack late endosomal proteins to release their viral genome into the host cell. This includes the main cholesterol transporter in LE/Lys, Niemann–Pick type C1 (NPC1), which serves as the entry factor for several filoviruses with Ebola virus using NPC1 in a cholesterol-independent manner (Carette et al, 2011; Cote et al, 2011). In addition, cholesterol accumulation in LE/Lys, using the pharma-cological NPC1 inhibitor U18666A, compromised fusion of the

influenza lipid envelope with late endosomal membranes (Musiol et al, 2013; Kuhnl et al, 2018; Schloer et al, 2019). Alike, U18666A and other drugs that cause LE/Lys cholesterol accumulation elicit antiviral activity (Sturley et al, 2020) and specifically, U18666A re-duced SARS-CoV-2 infection in Vero-E6 and Calu-3a cells (Schloer et al, 2020a). Thus, the disruption of cholesterol export from LE/Lys could serve as a pharmacological tool against SARS-CoV-2 (Schloer et al, 2020a; Sturley et al, 2020), but (Schloer et al, 2020a; Sturley et al, 2020) the cytotoxicity limits the therapeutical use of U18666A (Cenedella et al, 2004).

Alternatively, hydroxypropyl-beta-cyclodextrin serves as a cholesterol-depleting agent at the cell surface to reduce choles-terol accumulation in patients with NPC1 deficiency. This phar-maceutical impaired SARS-CoV-2 infection and replication and reduced expression of pro-inflammatory cytokines in human monocytes and Calu-3a cells (Bezerra et al, 2022). As hydroxypropyl-beta-cyclodextrin removes cholesterol from LE/Lys, the plasma membrane (Ottinger et al, 2014) and membranous organelles containing viral replication complexes (Bezerra et al, 2022), its mode of action to reduce SARS-CoV-2 infectivity remains to be clarified. Alike studies using MβCD (Lu et al, 2008), hydroxypropyl-beta-cyclodextrin did not alter ACE2 levels (Bezerra et al, 2022), in-dicating that these cholesterol-depleting cyclodextrins cause displacement of ACE2 from cholesterol-rich lipid rafts and con-sequently, reduced binding of the spike protein (Thorp & Gallagher, 2004; Li et al, 2007; Lu et al, 2008; Redondo-Morata et al, 2016; Kim et al, 2019; Fecchi et al, 2020; Cesar-Silva et al, 2022).

Miglustat inhibits glucosylceramide synthase, limiting ganglio-side buildup in LE/Lys in several lysosomal storage disorders, such as Tay-Sachs and Sandhoff disease (Mansouri et al, 2023). In NPC patients, glycosphingolipid levels in LE/Lys are also increased and miglustat is approved to ameliorate neuronal dysfunction in NPC1 deficiency (Lachmann et al, 2004). As miglustat is an iminosugar, this compound also inhibits α-glucosidases I and II in the ER, re-sponsible for early stages of glycoprotein N-linked oligosaccharide processing. The spike, envelope, and membrane proteins of SARS-CoV-2 are highly glycosylated proteins (Gong et al, 2021), and in SARS-CoV-2–infected cells, miglustat markedly decreased viral proteins, including the spike protein, and subsequent release of infectious viruses (Rajasekharan et al, 2021).

Other pharmacologicals with antiviral activities that cause LE/Lys-cholesterol accumulation include the antifungal itraconazole, a triazole derivative that blocks the synthesis of fungal ergosterol, but also binds and inhibits NPC1 (Trinh et al, 2017). Itraconazole has antiviral properties against a range of viruses in cell and mouse models (Schloer et al, 2019; Schloer et al, 2020b) and reduced SARS-CoV-2 infection in several cell lines (Liesenborghs et al, 2021; Schloer et al, 2021). However, a preclinical hamster model and a pilot clinical trial indicated this drug to lack clinical benefit (Liesenborghs et al, 2021).

In addition, inhibition of acid sphingomyelinase, which converts sphingomyelin to ceramide and phosphorylcholine, also causes cholesterol accumulation in LE/Lys (Lloyd-Evans et al, 2008; Schloer et al, 2020a). Small compounds that inhibit sphingomyelinase are clinically approved, well-tolerated, and widely used to treat a broad spectrum of pathological conditions (Kornhuber et al, 2010). Along these lines, fluvoxamine was beneficial for COVID-19 patients in two

recent studies (Kornhuber et al, 2022; Wen et al, 2022). A retrospective cohort study of almost 400,000 hospitalized COVID-19 patients suggested that prior use of antidepressant medications may reduce the likelihood of SARS-CoV-2 infection, hospitalization, and mortality. These associations between antidepressants and COVID-19 severity are most likely due to inhibition of acid sphingomyelinase (FIASMA) (Hoertel et al, 2022). Yet, fluvoxamine treatment of outpatients with mild to moderate COVID-19 did not improve the time to sustained recovery (Stewart et al, 2023), pointing at the need for further studies to evaluate the clinical effects of fluvoxamine in COVID-19.

Antidepressants including fluoxetine revealed reduced SARS-CoV-2 infection in Vero-E6 and Calu-3a cells and a decline in viral titers in murine lung tissue 3D ex vivo explants, which is supported by several retrospective and observational studies for fluoxetine and hydroxyzine over the course of COVID-19 (Schloer et al, 2021; Schloer et al, 2022). Strikingly, the combination of itraconazole or fluoxetine with remdesivir, a nucleotide analogue that inhibits the SARS-CoV-2 RNA polymerase, displayed stronger antiviral activities compared to monotherapy, indicating that targeting multiple regulatory pathways may be a valuable therapeutic strategy (Schloer et al, 2020a).

Other consequences associated with cholesterol accumulation in LE/Lys could potentiate capacity to reduce SARS-CoV-2 infection and propagation (Ballout et al, 2020). This includes the impaired proteolytic activity of cathepsin, which is critical for SARS-CoV-2 processing for cell entry in LE/Lys. The dysfunction of cathepsins appears related to cholesterol levels in this compartment, as clearance of accumulated cholesterol restored cathepsin B/L activity (Elrick & Lieberman, 2013; Ballout et al, 2020).

Blocked endolysosomal cholesterol efflux upon NPC1 inhibition causes cholesterol depletion in other cellular sites (Cubells et al, 2007; Musiol et al, 2013), such as the plasma membrane. Indeed, U18666A treatment or Rab7 inhibition, which also triggers LE/Lys-cholesterol accumulation (Meneses-Salas et al, 2020), reduced the number of released influenza virus progeny and lowered cholesterol in the viral envelope, both critical for the success of influenza infection (Musiol et al, 2013; Kuhnl et al, 2018). Yet, coronaviruses assemble at membranes of the ER Golgi intermediate compartment (ERGIC), followed by budding into the lumen and release via exocytosis of cargo vesicles (Stertz et al, 2007). As NPC1 inhibition also reduces cholesterol amounts in the ER, Golgi, and exocytic vesicles (Cubells et al, 2007; Reverter et al, 2014), one can speculate that blocked cholesterol export from LE/Lys may also lower cholesterol levels in ERGIC, with consequences for virus assembly and release.

Furthermore, with NPC1 inhibition lowering the amount of cholesterol-rich microdomains (Garver et al, 2002; Vainio et al, 2005; Cubells et al, 2007), one can envisage that lipid raft association of ACE2 could be compromised, with detrimental consequences for SARS-CoV-2 cell surface entry. In fact, type II pneumocytes, which are susceptible to SARS-CoV-2 infection and express ACE2 (Li et al, 2003; Hamming et al, 2004) depend on NPC1/2 to modulate the lipid composition of the pulmonary surfactant (Roszell et al, 2012; Rodriguez-Gil et al, 2019). Likewise, membrane proteases TMPRSS2 and ADAM17 are found predominantly in lipid rafts. As ADAM17 is elevated in NPC mutant cells, this might increase ACE2 shedding

and counteract TMPRSS2, adding further protection against viral cell surface binding (Ballout et al, 2020).

Also, elevated levels of the two oxysterols 7-ketocholesterol and 25-hydroxycholesterol, both with potent antiviral activities (Massey [2006]; Wang et al [2020b]; Zang et al [2020]; Mao et al [2022]), accumulate in NPC1 deficiency (Porter et al, 2010).

Several other factors of the endocytic machinery in LE/Lys linked to cholesterol and NPC1 function could also emerge as druggable antiviral targets. This includes endosomal sorting complex required for transport complexes (Du et al, 2012, 2013), late endosomal annexins (Musiol et al, 2013; Kuhnl et al, 2018; Enrich et al, 2021), and Rab-GTPases (Zang et al, 2020) and their regulators and effectors (Meneses-Salas et al, 2020), but also proteins that link cholesterol and calcium balance in LE/Lys (Lloyd-Evans et al, 2008; Enrich et al, 2021).

Finally, it should be noted that despite these many antiviral properties mediated by LE/Lys-cholesterol accumulation, NPC deficiency also lowers cholesterol levels in the ER, which is associated with elevated SREBP2 activity (Kristiana et al, 2008; Meneses-Salas et al, 2020). The latter may activate the NOD-, LRR-, and pyrin domain–containing protein 3 (NLRP3) inflammasome and contribute to inflammatory reactions (see below). Future studies will have to address these potentially undesired outcomes.

## The prognostic value of systemic cholesterol levels in COVID-19

Infectious diseases are long known to be associated with low LDL and HDL levels (Deniz et al, 2007; Pirillo et al, 2015), probably reflecting a higher cellular demand for cholesterol to support viral propagation. Thus, LDL-, HDL-cholesterol, and total serum cholesterol levels were reduced in COVID-19 patients (Lee et al, 2020; Kim et al, 2021a; Usenko et al, 2023) and associated with a higher risk for infection, a more severe disease course, and death (Wang et al, 2021; Zhao et al, 2021). Indicating recovery after infection, LDL-, HDL-, and total cholesterol increased 3–6 mo after discharge from hospital (Li et al, 2021a). Notably, not all studies support an association of circulating cholesterol levels with SARS-CoV-2 infection (Caterino et al, 2021; Kukla et al, 2021; Ruscica et al, 2021; Lavis et al, 2022).

## Statin use and COVID-19 severity and outcome

Statins are widely prescribed drugs effectively lowering systemic LDL-cholesterol levels (Grewal & Buechler, 2022) that also substantially reduce cholesterol levels at the plasma membrane. The latter disturbs the formation of cholesterol-rich microdomains, and lipophilic statins (e.g., lovastatin, simvastatin, pitavastatin, atorvastatin) probably displace ACE2 from cholesterol-rich lipid rafts, which reduces its ability to serve as SARS-CoV-2 receptor (Bakillah et al, 2022; Fiore et al, 2022). Furthermore, statins can increase ACE2 protein levels (Fiore et al, 2022; Teixeira et al, 2022), which may antagonize the harmful effects of ACE2 down-regulation that occur after ACE2-mediated uptake of SARS-CoV-2 (Fig 2). Upon viral entry through this route, ACE2 suppression causes overactivation of the renin angiotensin system, associated with vasoconstriction, inflammation, edema and fibrosis, all of which contributing to COVID-19 disease severity (De Spiegeleer et al, 2020; Silhol et al, 2020).

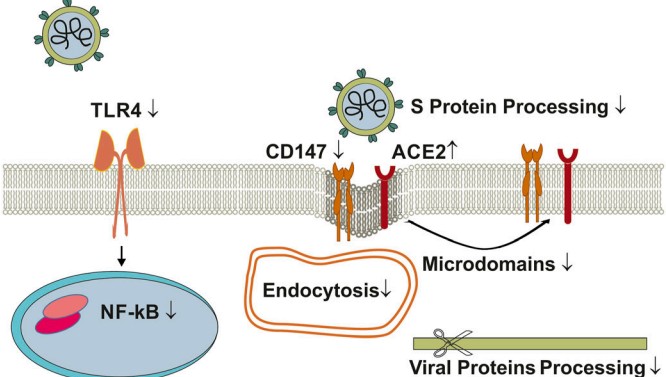

**Figure 2. Effect of statins on SARS-CoV-2 infection.**
Statin-induced ACE2 displacement from cholesterol-rich lipid rafts and lowering of CD147 plasma membrane levels interfere with viral fusion and/or viral entry via endocytosis. Statins prevent ACE2 down-regulation upon SARS-CoV-2 infection and inhibit NF-kB and TLR4-regulated inflammatory pathways. Abbreviations: ACE2, angiotensin-converting enzyme 2; CD147, cluster of differentiation 147; NF-kB, nuclear factor kappa-light-chain-enhancer of activated B cells; S protein, spike protein; TLR4, Toll-like receptor 4.

In addition, in THP-1 macrophages, statins down-regulated the cell surface levels of the alternative SARS-CoV-2 receptor CD147, likely because of its intracellular retention (Sasidhar et al, 2017; Bakillah et al, 2022). Statins also down-regulate TMPRSS2, thereby reducing the processing of the viral S2 subunit after docking to ACE2, ultimately lowering fusion of the virus with the plasma membrane (Fiore et al, 2022). Furthermore, the anti-inflammatory activities of statins (Zivkovic et al, 2023) inhibiting NF-kB and TLR4-induced pathways (Behl et al, 2022) (Fig 2) lowered inflammation of SARS-CoV-2–infected cell lines such as Calu-3a cells, lung tissues, human monocytes, and neutrophils (Behl et al, 2022; Fiore et al, 2022; Teixeira et al, 2022; Zivkovic et al, 2023). The proposed S protein binding to human TLR4 (Choudhury & Mukherjee, 2020) might also be counteracted by the anti-inflammatory action of statins (Choudhury & Mukherjee, 2020) (Fig 2).

Although experimental evidence is still lacking, several recent in silico studies have proposed statins binding to proteins other than 3-hydroxy-3-methylglutaryl-CoA reductase, such as viral proteins and SARS-CoV-2 receptors, possibly affecting virus entry and propagation (Reiner et al, 2020; Ghosh et al, 2022) (Fig 2).

On the other hand, statin use before COVID-1 hospitalization was not reported to protect against a fatal outcome (Hariyanto & Kurniawan, 2020). The loss of geranyl-pyrophosphate precursors compromising protein prenylation are considered to explain the anti-inflammatory activities of statins (Smaldone et al, 2009). However, this also reduces production of coenzyme Q10, which is required for mitochondrial electron transport that drives the generation of ATP. Hence, statin-mediated reduction of coenzyme Q10 availability in patients with existing mitochondrial dysfunction may amplify the risk of poor COVID-19 outcomes (Golomb et al, 2023).

Hence, further research is still required to better define those patients in SARS-CoV-2–infected cohorts with mixed comorbidities that benefit most from statin use (Deshpande et al, 2015; Chen et al, 2018; Pienkos et al, 2023). The differential activity

and pharmacokinetics of lipo- and hydrophilic statins also needs to be considered in this context.

## Other cholesterol-lowering drugs affecting COVID-19 disease

Besides statins, inhibitory antibodies against proprotein convertase subtilisin/kexin type 9 (PCSK9) are now used to treat hypercholesterinemia, increasing LDL receptor levels at the cell surface for enhanced LDL clearance (Sundararaman et al, 2021; Grewal & Buechler, 2022).

In addition, PCSK9 has several other functions, including its ability to promote ACE2 degradation (Essalmani et al, 2023), which might affect SARS-CoV-2 infectivity. On the other hand, in experimental models, PCSK9 inhibition reduced inflammation and improved survival in sepsis. SREBP2 as well as NF-kB, both transcription factors activated upon SARS-CoV-2 infection, can elevate PCSK9 expression (Grewal & Buechler, 2022; Elahi et al, 2023; Essalmani et al, 2023) and NF-kB or SREBP2 inhibitors normalized PCSK9 levels in PBMCs of COVID-19–infected patients (Lee et al, 2020).

Although not all studies support an association of PCSK9 expression and COVID-19 severity (Huang et al, 2021; Mester et al, 2023), the subcutaneous injection of evolocumab, a monoclonal inhibitory PCSK9 antibody, in patients with severe COVID-19 lowered the need for intubation and death. Although the lowering of circulating LDL-cholesterol levels may reduce cholesterol levels of peripheral cells and monocyte inflammation (Stiekema et al, 2021; Xie et al, 2022), the PCSK9 blockade improving COVID-19 outcomes cannot be easily explained by up-regulation of LDL receptor–mediated endocytosis and cellular cholesterol levels, a setting that would rather support COVID-19 infectivity. In fact, PCSK9 blockage reduced serum IL-6 levels, suggesting that lowering of circulating PCSK9 levels reduced inflammation, which may be partly due to increased ACE2 activity, which is associated with higher levels of the anti-inflammatory angiotensin 1–7 (Essalmani et al, 2023; Navarese et al, 2023). Along these lines, PCSK9-deficient mice did not display increased liver cholesterol and bile acid levels and may not cause cholesterol accumulation in different cells and tissues (Parker et al, 2013).

Ezetimibe blocks NPC1L1-mediated intestinal cholesterol absorption, thereby lowering systemic cholesterol levels. In addition, ezetimibe inhibits hepatic NPC1L1, up-regulates hepatic LDL receptor levels, and promotes biliary cholesterol excretion (Sudhop et al, 2009; Pramfalk et al, 2011; Jocher et al, 2022; Nejat et al, 2023). In several cell models, ezetimibe impaired viral entry (Chen et al, 2021), which correlates with reduced hospitalization of COVID-19 patients (Israel et al, 2021).

Pantethine, the dimer of pantetheine, an amid analogue of pantothenic acid (vitamin B5) inhibits fatty acid and cholesterol synthesis and lowered SARS-CoV-2 infection. This was accompanied by a greatly reduced expression of viral proteins and cellular release of viral particles. Antiviral activities of pantethine include down-regulation of host proteins needed for viral entry such as TMPRSS2 and inflammatory genes (Abou-Hamdan et al, 2023). Higher vitamin B5 intake correlated with a lower incidence of COVID-19 (Darand et al, 2022) and molecular docking studies

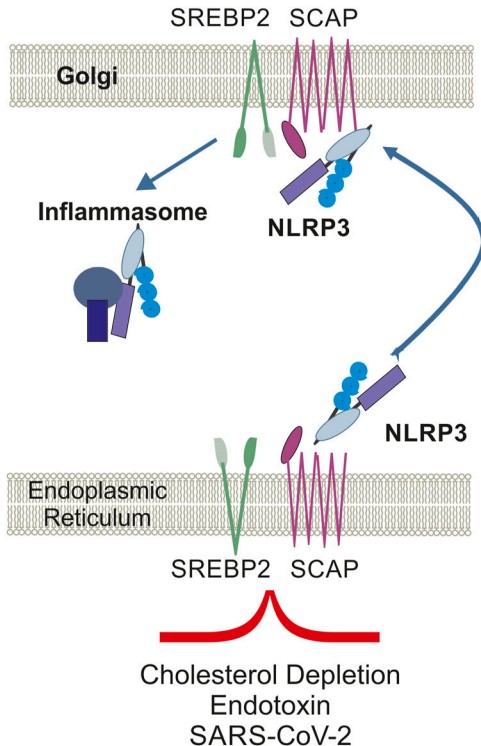

**Figure 3. SREBP2 activation is related to inflammation in PBMCs of COVID-19 patients.**
SCAP-mediated escort of SREBP2 from the ER to the Golgi apparatus is required for the translocation of NLRP3 to the Golgi, enabling formation and activation of the inflammasome. Abbreviations: NF-kB, nuclear factor kappa-light-chain-enhancer of activated B cells; NLRP3, NOD-, LRR-, and pyrin domain–containing protein 3; PBMCs, peripheral blood mononuclear cells; SARS-CoV-2, severe acute respiratory syndrome coronavirus 2; SCAP, SREBP2 cleavage–activating protein; SREBP2, sterol regulatory element–binding protein 2.

suggested pantethine as a possible inhibitor of the main viral protease $M^{pro}$ (Verma et al, 2022).

### SREBP2, multiple roles for the master regulator of sterol synthesis in COVID-19

The inappropriate innate immune response to SARS-CoV-2 is triggered by the activation of the NLRP3 inflammasome. As NLRP3 inhibition reduced COVID-19–like hyperinflammation and pathology in preclinical models, this has potential to treat severe SARS-CoV-2 complications (Diarimalala et al, 2023; Potere et al, 2023).

Upon statin-induced cholesterol depletion, NLRP3 activation and translocation to the Golgi apparatus occurs in an SREBP2-dependent manner, whereas cholesterol supplementation or SREBP2 blockage inhibited NLRP3 inflammasome activation and protected from inflammation (Guo et al, 2018).

Interestingly, despite elevated transcriptional activity of SREBP2 in COVID-19 disease (Swanson et al, 2019), a disconnect between SREBP2 activation in inflammation with cholesterol homeostasis was observed (Guo et al, 2018; Lee et al, 2020). Subsequent studies identified NF-kB inhibition to block SREBP2 activity in PBMCs of COVID-19 patients to reduce production of inflammatory cytokines. This inflammatory activity of SREBP2 was then validated in a rodent

model, with pharmacological SREBP2 inhibition effectively lowering inflammation in sepsis (Lee et al, 2020).

Elevated TNF levels in COVID-19 patients (Han et al, 2020) can activate SREBP2, which induces interferon response and inflammatory gene expression in macrophages (Fig 3). Likewise, atorvastatin enhanced SREBP2-mediated up-regulation of inflammatory genes (Kusnadi et al, 2019). However, SARS-CoV-2–infected lung cancer cell lines A549 and Calu-3a revealed down-regulation of SREBP2 (Gomez Marti et al, 2021), indicating cell-dependent and differential outcome of SARS-CoV-2 infection on cholesterol homeostasis and SREBP2 activity.

## Conclusion

In this review, we described several key roles how cholesterol at the cell surface and in endolysosomes influences the efficacy of SARS-CoV-2 entry, providing opportunities for drugs targeting cholesterol-sensitive mechanism in these locations to reduce SARS-CoV-2 infectivity and COVID-19 disease severity. It would have gone beyond the scope of this review to cover the influence of cholesterol on many other aspects of the viral cycle, such as replication, assembly, and viral release, and we refer the reader to excellent review articles that cover cholesterol-sensitive mechanisms not only during SARS-CoV-2 surface recognition and cell entry but also viral replication, assembly, and release (Ballout et al, 2020; Fecchi et al, 2020; Tang et al, 2021b; Glitscher & Hildt, 2021; Palacios-Rapalo et al, 2021; Barrantes, 2022; Cesar-Silva et al, 2022; Kowalska et al, 2022; Wang et al, 2022, 2023; Ahmad et al, 2023).

Alike many other viruses, SARS-CoV-2 takes advantage of cellular cholesterol supply to gain entry and propagate in the host cell. Elevated cellular cholesterol levels support the generation of SARS-CoV-2 progeny for subsequent viral infections. These observations correlate with low serum cholesterol levels, disease severity, and death in COVID-19 patients. In many studies, patients using statins appear to experience less severe disease. Although statins may instigate SREBP2-dependent NLRP3 inflammasome activation in PBMCs, this does not appear to impact significantly in the various statin-related epidemiological studies in larger COVID-19 cohorts. The promising beneficial effects of inhibitory PCSK9 antibodies and other cholesterol-depleting drugs, which protected from viral infection in preclinical studies, have yet to be validated in patient cohorts. FDA-approved drugs that cause cholesterol accumulation in LE/Lys through NPC1 inhibition also provide opportunity to combat SARS-CoV-2 infectivity. With multiple cholesterol-sensitive events influencing SARS-CoV-2 infection, propagation, and release, further efforts to develop strategies that interfere with the availability of cholesterol for critical steps in the infectious cycle may be of benefit in the therapy of COVID-19. As mentioned earlier, age is a critical risk factor for COVID-19 severity. This correlates with cholesterol levels commonly increasing with age (Cutler et al, 2004; Wang et al, 2023) and disease (Xiong et al, 2008; Rudajev & Novotny, 2022). On the other hand, children have comparatively lower cholesterol levels (Cutler et al, 2004) and a better outcome than the elderly when infected with SARS-CoV-2 (Kang et al, 2022). This age-related correlation between cholesterol levels and disease severity

supports strategies aiming to lower cholesterol in the elderly to reduce COVID-19 severity.

Cholesterol plays a crucial role during multiple steps in the life cycle of almost all viruses. Thus, manipulating cholesterol homeostasis could potentially serve as a broad-spectrum antiviral strategy. As outlined in this review, rather than reducing cholesterol levels globally, the key to cholesterol-related antiviral strategies may require disruption of cholesterol handling in specific cellular organelles where it is needed for viral entry, replication, and propagation (Glitscher & Hildt, 2021). Endolysosomal cholesterol homeostasis has been recognized as a potential target for antiviral treatments against a variety of viruses, including SARS-CoV-2, but also influenza A and Ebola viruses. In the case of Ebola, NPC1 is hijacked by the virus to cross the endolysosomal membrane for cell entry. Hence, drugs interfering with NPC1 activity and endolysosomal cholesterol levels such as itraconazole and fluoxetine have potential to deliver broad antiviral effects (Glitscher & Hildt, 2021; Kummer et al, 2022). Although the availability of these tools offer promising treatment options against viral infections, further research is still needed to fully understand the impact of interfering with cholesterol transport in cells and to assess adverse effects of these drugs (Glitscher & Hildt, 2021).

## Supplementary Information

### Author Contributions

T Grewal: writing—review and editing.
MKL Nguyen: writing—review and editing.
C Buechler: writing—original draft and writing—review and editing.

### Conflict of Interest Statement

The authors declare that they have no conflict of interest.

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
