## [Reviewer comments · Life Science Alliance]

Life Science Alliance

Cholesterol and COVID-19 - therapeutic opportunities at the host/virus interface during cell entry

Thomas Grewal, Mai Khanh Linh Nguyen, and Christa Buechler

DOI: <https://doi.org/10.26508/lsa.202302453>

Corresponding author(s): Christa Buechler, University Hospital Regensburg

Review Timeline:

Submission Date:	2023-10-23
Editorial Decision:	2023-12-18
Revision Received:	2024-01-17
Editorial Decision:	2024-02-05
Revision Received:	2024-02-12
Accepted:	2024-02-13

Transaction Report:

December 18, 2023

Re: Life Science Alliance manuscript #LSA-2023-02453-T

Prof. Christa Buechler
Regensburg University Hospital
Franz Josef Strauss Allee 11
Regensburg 93042
GERMANY

Dear Dr. Buechler,

Thank you for submitting your manuscript entitled "Cholesterol and COVID-19 - therapeutic opportunities at the host/virus interface during cell entry" to Life Science Alliance. The manuscript was assessed by expert reviewers, whose comments are appended to this letter. We invite you to submit a revised manuscript addressing the Reviewer comments.

Thank you for this interesting contribution to Life Science Alliance. We are looking forward to receiving your revised manuscript.

Sincerely,

B. MANUSCRIPT ORGANIZATION AND FORMATTING:

Reviewer #1 (Comments to the Authors (Required)):

Grewal et al. present a review on the therapeutic potential of cholesterol modifying drugs in the treatment of COVID. Cholesterol has become well recognized culprit in SARS-CoV-viral entry. The review is well written, covers most of the key points of cholesterol as a target of COVID19 treatment, and it is timely. I recommend publication. A few mostly minor points are provided below.

Major points:

Page 5 paragraph 1 make sure distinguish blood cholesterol from tissue cholesterol, they are not necessarily the same.

Page 5 paragraph 3 describes removing cholesterol with MBCD and the effects on rafts, this has also been done with ApoE[1]. ApoE is an endogenous cholesterol transport molecule that regulates uptake and export of cholesterol from peripheral tissue. The ApoE is important because the clinical outcomes of COVID19 are linked to isotope of ApoE[2,3]. A short discussion on ApoE could add support for the likelihood that clinical outcomes are likely to improve with MBCD treatments or other cholesterol lowering drugs.

Most the references to lipid rafts are in the era of detergent resistant membranes. More recent studies have used super resolution imaging to look at lipid rafts and ACE2 in the membrane of intact cells and even whole tissue. For example, in a diabetic mouse model, cholesterol in the tissue goes up with age and disease and this correlates with ACE2 associating with cholesterol rich GM1 lipids in whole tissue slices of fixed mouse lung[1,4]. For nanodomain regulation, PIP2 is also important. ACE2 and the spike protein traffics to PIP2 clusters[1,5] in intact cells, and hydroxychloroquine disruption the association with GM1 and PIP2 in multiple cells types[6].

Pg 11 p 4. The argument for using PCSK9 in COVID-19 is not well justified to this reviewer. Increasing LDL receptor will increase cholesterol uptake into cells[7]. Increasing cholesterol uptake should be harmful to treating COVID-19, not helpful. It is true that some individuals have high blood cholesterol because their cells don't take it up very well (e.g. homozygous apoE2)-- they are likely deficient in their tissue. For those cases, correcting the uptake by enhancing LDL receptor will lower the blood cholesterol level and be helpful. But for COVID-19, low tissue cholesterol is the goal for treating COVID-19, not low blood cholesterol, as this decreases inflammation and the potential for a cytokine storm.

Minor points

Might be worth a mention of avasimibe (ACAT inhibitor) in "other mechanism". It is yet another strategy for inhibiting endocytosis of the virus[8]. ACAT is responsible for uptake of cholesterol into internal stores and appears to block SARS-Cov-2 infection.

Pg 9 p3, the reference to high levels of ACE2 (Hamming 2004 and Li 2003) don't show actual levels of ACE2 dictate infectivity in a physiological setting. Paul McCray showed protein is actually higher in young children, yet children have least severity of the disease (see 2020 EBioMedicine Fig. 4 e). Over expressed ACE2 in the cultured studies likely saturates Gm1 domains which will increase endocytosis, as discussed in the Li 2004 reference. Please clarify or at least provide a reference where high protein levels correlate with higher infectivity in physiological tissue.

Page 5 hydroxychloroquine, reports toxicity, but all the large studies (>500 participants) were done at 2x or more the FDA recommended dose of 200-400 mg per day[9]. Half-life of HCQ is 123.5 days in plasma, so the daily high concentrations may be the cause of the toxicity, not that the drug itself when used as recommended by the FDA. This information is important for those who use the drug for Niemann-Pick and as a prophylactic in malaria treatment.
https://www.accessdata.fda.gov/drugsatfda_docs/label/2021/009768s053lbl.pdf

Suggestions:

Age is a major factor in COVID-19 severity. When normalized to protein, cholesterol increases with age[1] (figure 3a) and [10] (figure 1c) and disease[11,12]. While this is still mostly a correlation, the favorable outcomes of children for COVID19 and the

fact that they have lower tissue cholesterol, is a promising example of what a cholesterol lowering drug should be able to do for aged and diseased persons. Cholesterol almost certainly plays a role in the difference severity between adults and children.

Pg5 p5 Probably might be worth mentioning that more recent variants seems to have evolved away from TMPRSS[13], relying more on cholesterol and the endocytic pathway. And more recent studies suggests the early variants of SARS-CoV-2 probably depended more on cholesterol then was originally suggested with the TMPRSS2 sutdies[1,4,6,14].

References cited

1. Wang H, Yuan Z, Pavel MA, Jablonski SM, Jablonski J, Hobson R, et al. The role of high cholesterol in SARS-CoV-2 infectivity. *J Biol Chem.* 2023;299: 104763. doi:10.1016/j.jbc.2023.104763
2. Kuo C-L, Pilling LC, Atkins JL, Masoli JAH, Delgado J, Kuchel GA, et al. APOE e4 genotype predicts severe COVID-19 in the UK Biobank community cohort. *Journals Gerontol Ser A.* 2020; 1-29. doi:10.1093/gerona/glaa131
3. Gkouskou K, Vasilogiannakopoulou T, Andreakos E, Davanos N, Gazouli M, Sanoudou D, et al. COVID-19 enters the expanding network of apolipoprotein E4-related pathologies. *Redox Biol.* 2021;41. doi:10.1016/j.redox.2021.101938
4. Miao L, Yan C, Chen Y, Zhou W, Zhou X, Qiao Q, et al. SIM imaging resolves endocytosis of SARS-CoV-2 spike RBD in living cells. *Cell Chem Biol.* 2023;30: 248-260.e4. doi:10.1016/j.chembiol.2023.02.001
5. Raut P, Waters H, Zimmerberg J, Obeng B, Gosse J, Hess ST. Localization-based super-resolution microscopy reveals relationship between SARS-CoV2 spike and phosphatidylinositol (4,5): biphosphate. 2022;11965: 10. doi:10.1117/12.2613460
6. Yuan Z, Pavel MA, Wang H, Kwachukwu JC, Mediouni S, Jablonski JA, et al. Hydroxychloroquine blocks SARS-CoV-2 entry into the endocytic pathway in mammalian cell culture. *Commun Biol.* 2022;5: 958. doi:10.1038/s42003-022-03841-8
7. Hansen SB, Wang H. The shared role of cholesterol in neuronal and peripheral inflammation. *Pharmacol Ther.* 2023;249: 108486. doi:10.1016/j.pharmthera.2023.108486
8. Wing PA, Schmidt NM, Peters R, Erdmann M, Brown R, Wang H, et al. An ACAT inhibitor suppresses SARS-CoV-2 replication and boosts antiviral T cell activity. Pfaender S, editor. *PLOS Pathog.* 2023;19: e1011323. doi:10.1371/journal.ppat.1011323
9. Axfors C, Schmitt AM, Janiaud P, van't Hooft J, Abd-Elsalam S, Abdo EF, et al. Mortality outcomes with hydroxychloroquine and chloroquine in COVID-19 from an international collaborative meta-analysis of randomized trials. *Nat Commun.* 2021;12: 1-13. doi:10.1038/s41467-021-22446-z
10. Cutler RG, Kelly J, Storie K, Pedersen WA, Tammara A, Hatanpaa K, et al. Involvement of oxidative stress-induced abnormalities in ceramide and cholesterol metabolism in brain aging and Alzheimer's disease. *Proc Natl Acad Sci U S A.* 2004;101: 2070-2075. doi:10.1073/pnas.0305799101
11. Rudajev V, Novotny J. Cholesterol as a key player in amyloid β -mediated toxicity in Alzheimer's disease. *Front Mol Neurosci.* 2022;15: 1-23. doi:10.3389/fnmol.2022.937056
12. Xiong H, Callaghan D, Jones A, Walker DG, Lue LF, Beach TG, et al. Cholesterol retention in Alzheimer's brain is responsible for high β - and γ -secretase activities and A β production. *Neurobiol Dis.* 2008;29: 422-437. doi:10.1016/j.nbd.2007.10.005
13. Meng B, Abdullahi A, Ferreira IATM, Goonawardane N, Saito A, Kimura I, et al. Altered TMPRSS2 usage by SARS-CoV-2 Omicron impacts infectivity and fusogenicity. *Nature.* 2022;603: 706-714. doi:10.1038/s41586-022-04474-x
14. Kreutzberger AJB, Sanyal A, Saminathan A, Bloyet L-M, Stumpf S, Liu Z, et al. SARS-CoV-2 requires acidic pH to infect cells. *Proc Natl Acad Sci.* 2022;119: 1-12. doi:10.1073/pnas.2209514119

Reviewer #2 (Comments to the Authors (Required)):

A review should aim at covering the subject matter with conciseness, fair bibliographic coverage and discussion that summarises the state-of-the-art of the field and its future directions. This reviews encompasses a rather limited analysis of the field and its bibliographic coverage is also rather limited, ignoring various recent contributions with detailed mechanistic hypothesis on the influence of cholesterol on SARS-CoV-1 cell-surface recognition, endocytosis and subsequent steps. The review is not suitable in its current state.

Reviewer #3 (Comments to the Authors (Required)):

In the manuscript entitled "Cholesterol and COVID-19 - therapeutic opportunities at the host/virus interface during cell entry", the authors provide a summary of the current knowledge in the field of cholesterol and COVID-19 and the current achievements in drug repurposing. Targeting host factors that support virus entry, replication and/or propagation are considered as highly effective, helping to lower viral titers and to ameliorate COVID-19 severity. As many enveloped viruses, such as SARS-CoV-2, are hijacking the cellular transport machinery and fuse with the endolysosomal membrane to get access to the cell, pharmaceuticals that impede viral fusion or modulate lipid composition of the respective cellular compartments are very promising candidates for antiviral therapies.

I want to thank the authors to bring this timely topic up, and I really like the overall structure of the manuscript. I have only a few suggestions for the authors for improving the manuscript.

1) The antiviral potential of NPC-1 inhibition and fluoxetine was recently evaluated using a more advanced human and murine 3D model (PMID: 36000328). I suggest to include the reference to bridge the gap between cell culture, animal models and the human system.

2) The clinical use of the two antidepressants with high FIASMA activity has been reported to reduce SARS-CoV-2 infections and COVID-19-related mortality in inpatients, and may be appropriate for prophylaxis and/or COVID-19 therapy for outpatients or inpatients (PMID: 36233753). This would strengthen the translational aspect of the comprised research. In this context, the following reference might also be of interest (PMID: 37976072).

3) Interestingly, pharmacological induced endolysosomal cholesterol imbalance through the clinically licensed drugs itraconazole and fluoxetine were reported to impair another virus with pandemic potential: Ebola virus (PMID: 34919035). As Ebola virus entry requires the cholesterol transporter Niemann-Pick C1 (PMID: 26771495), the use of these cholesterol affecting drugs might be promising for future pandemic management of other enveloped viruses that use the same entry route. The authors should include a small paragraph in the conclusions to outline the broad antiviral effect of targeting cellular cholesterol levels.

The presented manuscript is well-written and covers a timely topic. I hope that the authors can provide a revised manuscript including my suggestions.

Reviewer #1 (Comments to the Authors (Required)):

Grewal et al. present a review on the therapeutic potential of cholesterol modifying drugs in the treatment of COVID. Cholesterol has become well recognized culprit in SARS-CoV-viral entry. The review is well written, covers most of the key points of cholesterol as a target of COVID19 treatment, and it is timely. I recommend publication. A few mostly minor points are provided below.

We thank the Reviewer for the very positive, kind and extremely helpful comments on our review article.

Major points:

1. *Page 5 paragraph 1 make sure distinguish blood cholesterol from tissue cholesterol, they are not necessarily the same.*

We apologize for this oversight, and in the revised manuscript, this was clarified accordingly (Page 5, paragraph 1 of the chapter “Role of cholesterol at the plasma membrane for SARS-CoV-2 infection”

2. *Page 5 paragraph 3 describes removing cholesterol with MBCD and the effects on rafts, this has also been done with ApoE[1]. ApoE is an endogenous cholesterol transport molecule that regulates uptake and export of cholesterol from peripheral tissue. The ApoE is important because the clinical outcomes of COVID19 are linked to isotope of ApoE[2,3]. A short discussion on ApoE could add support for the likelihood that clinical outcomes are likely to improve with MBCD treatments or other cholesterol lowering drugs.*

We thank the Reviewer for this insight and absolutely agree that this aspect should be included in this review article. We have therefore added a brief description, including four new references, on ApoE-related findings relevant for COVID-19 on page 6, 3rd para. The reference list was corrected accordingly:

‘Besides using the cholesterol-depleting agent M□CD, apolipoprotein E (ApoE) is an endogenous cholesterol transport protein that in the non-lipidated form removes cholesterol, while the lipidated form is a physiological cholesterol donor for cells. Most relevant for COVID-19, lipidated ApoE induced ACE2 movement to GM1 rafts and enhanced virus infection (Wang *et al*, 2023). Three apoE isoforms exist in humans, apoE2, apoE3 and apoE4, and it is in this order that these genotypes are associated with increasing LDL-cholesterol levels and coronary artery disease (Tudorache *et al*, 2017). Homozygous apoE4 carriers had an increased risk of severe COVID-19 infection, independent of pre-existing comorbidities (Kuo *et al*, 2020). As the apoE4 genotype is associated with higher tissue cholesterol levels, oxidative stress and inflammation, this may all contribute to greater susceptibility to SARS-CoV-2 infection and COVID-19 severity (Gkouskou *et al*, 2021).’

3. Most the references to lipid rafts are in the era of detergent resistant membranes. More recent studies have used super resolution imaging to look at lipid rafts and ACE2 in the membrane of intact cells and even whole tissue. For example, in a diabetic mouse model, cholesterol in the tissue goes up with age and disease and this correlates with ACE2 associating with cholesterol rich GM1 lipids in whole tissue slices of fixed mouse lung[1,4]. For nanodomain regulation, PIP2 is also important. ACE2 and the spike protein traffics to PIP2 clusters[1,5] in intact cells, and hydroxychloroquine disruption the association with GM1 and PIP2 in multiple cells types(Stewart et al).

We thank the Reviewer for these helpful comments, which have been incorporated into the final paragraph on page 5 and the first para on page 6. Three new references were added and the reference list was corrected accordingly.

Page 5, bottom para: 'These specialized cholesterol-rich microdomains are heterogenous in terms of their lipid and protein content, often conferring different functions. For instance, lipid rafts enriched in either saturated ceramide-containing glycolipids such as monosialotetrahexosylganglioside 1 (GM1) or phosphatidylinositol 4,5 bisphosphate (PIP2) containing lipid clusters exist and movement of GM1 localized membrane proteins to PIP2 lipid clusters upon localized changes in membrane cholesterol levels may affect their biologic function (Yuan & Hansen, 2023).'

Page 6, first para: 'Interestingly, in a diabetic mouse model characterized by elevated cholesterol with increased age and disease, ACE2 was predominantly found in cholesterol-rich GM1 membrane structures that facilitate endocytosis. and cholesterol depletion caused ACE2 to move from GM1 lipids to PIP2 lipid domains, thereby decreasing virus uptake and infectivity (Wang et al., 2023). Super-resolution microscopy also revealed a relationship between the Spike protein and PIP2 clusters (Raut et al, 2022). In addition, hydroxychloroquine altered the association of ACE with GM1- and PIP2-containing microdomain clusters in cells with high and low cholesterol, respectively (Yuan et al, 2022). Also, avasimibe-mediated inhibition of Acyl-CoA:cholesterol acyltransferase (ACAT)-mediated cholesterol esterification disrupted ACE2 association with GM1 lipid rafts and inhibited SARS-CoV-2 pseudoparticle infection (Wing et al, 2023).'

4. Pg 11 p 4. The argument for using PCSK9 in COVID-19 is not well justified to this reviewer. Increasing LDL receptor will increase cholesterol uptake into cells[7]. Increasing cholesterol uptake should be harmful to treating COVID-19, not helpful. It is true that some individuals have high blood cholesterol because their cells don't take it up very well (e.g. homozygous apoE2)--they are likely deficient in their tissue. For those cases, correcting the uptake by enhancing LDL receptor will lower the blood cholesterol level and be helpful. But for COVID-19, low tissue cholesterol is the goal for treating COVID-19, not low blood cholesterol, as this decreases inflammation and the potential for a cytokine storm.

We absolutely agree with the Reviewer that PCSK9 inhibition causing reduced COVID severity cannot easily be explained by the upregulation of LDL-cholesterol uptake. Alternatively, as also suggested by this Reviewer, reduced inflammation upon PCSK9 inhibition could be the underlying cause for less severe disease outcome. In the revised manuscript, this is now discussed in more detail on page 14, 2nd para:

‘Although the lowering of circulating LDL-cholesterol levels may reduce cholesterol levels of peripheral cells and monocyte inflammation (Stiekema *et al*, 2021; Xie *et al*, 2022), the PCSK9 blockade improving COVID-19 outcomes cannot be easily explained by upregulation of LDL receptor-mediated endocytosis and cellular cholesterol levels, a setting that would rather support COVID-19 infectivity. In fact, PCSK9 blockage reduced serum IL-6 levels, suggesting that lowering of circulating PCSK9 levels reduced inflammation, which may be partly due to increased ACE2 activity, which is associated with higher levels of the anti-inflammatory angiotensin 1-7 (Essalmani *et al*, 2023; Navarese *et al*, 2023). Along these lines, PCSK9-deficient mice did not display increased liver cholesterol and bile acid levels and may not cause cholesterol accumulation in different cells and tissues (Parker *et al*, 2013).’

Minor points

5. *Might be worth a mention of avasimibe (ACAT inhibitor) in "other mechanism". It is yet another strategy for inhibiting endocytosis of the virus[8]. ACAT is responsible for uptake of cholesterol into internal stores and appears to block SARS-Cov-2 infection.*

We thank the Reviewer for this valuable suggestion. The impact of ACAT inhibitor on SARS-CoV2 infection, including one new reference, is now provided at the end of the first para on page 6, 2nd para (see also Point 3 above). The reference list was corrected accordingly:

‘Also, avasimibe-mediated inhibition of Acyl-CoA:cholesterol acyltransferase (ACAT)-mediated cholesterol esterification disrupted ACE2 association with GM1 lipid rafts and inhibited SARS-CoV-2 pseudoparticle infection (Wing *et al.*, 2023).’

6. *Pg 9 p3, the reference to high levels of ACE2 (Hamming 2004 and li 2003) don't show actual levels young children, yet children have least severity of the disease (see 2020 EBioMedicine Fig. 4 e). Over expressed ACE2 in the cultured studies likely saturates Gm1 domains which will increase endocytosis, as discussed in the Li 2004 reference. Please clarify or at least provide a reference where high protein levels correlate with higher infectivity in physiological tissue.*

This was clarified at the end of page 3 and corrected at page 11

We thank the Reviewer for these important observations. In the revised manuscript, the complex relationship between ACE2 protein levels and infectivity has been described in more detail on Page 3-

4. The impact of cholesterol on ACE2 localization and activity in GM1 domains has been clarified on page 6, 1st para (see point 3) and was corrected at page 11.

Page 3-4: ‘In the respiratory system, ACE2 was not detectable or expressed at low levels in some cells but was present in many other cells and tissues such as enterocytes, gallbladder, cardiomyocytes, ductal cells and vasculature, indicating that high ACE2 expression is not an indicator of increased susceptibility to infection (Hikmet *et al*, 2020). Consistent with these findings, only a relatively small number of ACE2-positive cells were found in the human respiratory tract. Further indicating that ACE2 protein levels per se do not determine susceptibility to COVID-19 infection, increased ACE2 protein levels were found in individuals at lower risk of severe COVID-19, such as children and healthy controls (Ortiz *et al*, 2020).’

Page 11: ‘In fact, type II pneumocytes, which are susceptible to SARS-CoV-2 infection and express ACE2. ‘

7. Page 5 hydroxychloroquine, reports toxicity, but all the large studies (>500 participants) were done at 2x or more the FDA recommended dose of 200-400 mg per day[9]. Half-life of HCQ is 123.5 days in plasma, so the daily high concentrations may be the cause of the toxicity, not that the drug itself when used as recommended by the FDA. This information is important for those who use the drug for Niemann-Pick and as a prophylactic in malaria treatment.

https://www.accessdata.fda.gov/drugsatfda_docs/label/2021/009768s053lbl.pdf

We thank the Reviewer for this important information and in the revised manuscript, we have added additional information on the use of hydroxychloroquine on page 4 (bottom para). This included one new reference and the reference list was corrected accordingly:

‘Of note, in several clinical studies with >500 participants, twice or more the FDA recommended doses of hydroxychloroquine were used, yet these studies did not provide conclusive evidence of the effectiveness of hydroxychloroquine in treating COVID-19 (Axfors *et al*, 2021; Kumar *et al*, 2021). Yet, hydroxychloroquine, as well as chloroquine, can severely compromise vital LE/Lys functions, and very high doses can lead to cell death and strong side-effects and toxicity profiles.’

Suggestions:

8. Age is a major factor in COVID-19 severity. When normalized to protein, cholesterol increases with age[1] (figure 3a) and [10] (figure 1c) and disease[11,12]. While this is still mostly a correlation, the favorable outcomes of children for COVID19 and the fact that they have lower tissue cholesterol, is a promising example of what a cholesterol lowering drug should be able to do for aged and diseased persons. Cholesterol almost certainly plays a role in the difference severity between adults and children.

We agree with the Reviewer's suggestion and added a short section at the end of the 1st para in the Conclusions on page 16. This included the addition of three new references and the reference list was corrected accordingly:

‘As mentioned earlier, age is a critical risk factor for COVID-19 severity. This correlates with cholesterol levels commonly increasing with age (Cutler *et al*, 2004; Wang *et al.*, 2023) and disease (Rudajev & Novotny, 2022; Xiong *et al*, 2008). On the other hand, children have comparatively lower cholesterol levels (Cutler *et al.*, 2004) and a better outcome than the elderly when infected with SARS-CoV-2 (Kang *et al*, 2022). This age-related correlation between cholesterol levels and disease severity supports strategies aiming to lower cholesterol in the elderly to reduce COVID-19 severity.’

9. Pg5 p5 Probably might be worth mentioning that more recent variants seems to have evolved away from TMPRSS[13], relying more on cholesterol and the endocytic pathway. And more recent studies suggests the early variants of SARS-CoV-2 probably depended more on cholesterol then was originally suggested with the TMPRSS2 sutdies[1,4,6,14].

We thank the Reviewer for this information on recent literature and have added the following short section to the third para on page 8. Two new references were added to this section and the reference list was corrected accordingly:

‘Remarkably, more recent variants of SARS-CoV-2, such as the Omicron BA.1 variant, displayed a lower efficiency in using the cellular protease TMPRSS2 and relied more on cell entry through endocytic pathways that are sensitive to cholesterol (Meng *et al*, 2022). A recent study highlighting the requirement of an acidic endosomal environment for early variants of SARS-CoV-2 also point at the potential to target late endosomal cholesterol (Kreutzberger *et al*, 2022). Hence, drugs targeting endocytic pathways and late endosomal cholesterol may be effective for therapy of patients infected with both recent and earlier variants. ‘

References cited

- 1. Wang H, Yuan Z, Pavel MA, Jablonski SM, Jablonski J, Hobson R, et al. The role of high cholesterol in SARS-CoV-2 infectivity. J Biol Chem. 2023;299: 104763. doi:10.1016/j.jbc.2023.104763*
- 2. Kuo C-L, Pilling LC, Atkins JL, Masoli JAH, Delgado J, Kuchel GA, et al. APOE e4 genotype predicts severe COVID-19 in the UK Biobank community cohort. Journals Gerontol Ser A. 2020; 1-29. doi:10.1093/gerona/glaa131*
- 3. Gkouskou K, Vasilogiannakopoulou T, Andreakos E, Davanos N, Gazouli M, Sanoudou D, et al. COVID-19 enters the expanding network of apolipoprotein E4-related pathologies. Redox Biol. 2021;41. doi:10.1016/j.redox.2021.101938*

4. Miao L, Yan C, Chen Y, Zhou W, Zhou X, Qiao Q, et al. SIM imaging resolves endocytosis of SARS-CoV-2 spike RBD in living cells. *Cell Chem Biol.* 2023;30: 248-260.e4. doi:10.1016/j.chembiol.2023.02.001
5. Raut P, Waters H, Zimmerberg J, Obeng B, Gosse J, Hess ST. Localization-based super-resolution microscopy reveals relationship between SARS-CoV2 spike and phosphatidylinositol (4,5)-biphosphate. 2022;11965: 10. doi:10.1117/12.2613460
6. Yuan Z, Pavel MA, Wang H, Kwachukwu JC, Mediouni S, Jablonski JA, et al. Hydroxychloroquine blocks SARS-CoV-2 entry into the endocytic pathway in mammalian cell culture. *Commun Biol.* 2022;5: 958. doi:10.1038/s42003-022-03841-8
7. Hansen SB, Wang H. The shared role of cholesterol in neuronal and peripheral inflammation. *Pharmacol Ther.* 2023;249: 108486. doi:10.1016/j.pharmthera.2023.108486
8. Wing PA, Schmidt NM, Peters R, Erdmann M, Brown R, Wang H, et al. An ACAT inhibitor suppresses SARS-CoV-2 replication and boosts antiviral T cell activity. Pfaender S, editor. *PLOS Pathog.* 2023;19: e1011323. doi:10.1371/journal.ppat.1011323
9. Axfors C, Schmitt AM, Janiaud P, van't Hooft J, Abd-Elsalam S, Abdo EF, et al. Mortality outcomes with hydroxychloroquine and chloroquine in COVID-19 from an international collaborative meta-analysis of randomized trials. *Nat Commun.* 2021;12: 1-13. doi:10.1038/s41467-021-22446-z
10. Cutler RG, Kelly J, Storie K, Pedersen WA, Tammara A, Hatanpaa K, et al. Involvement of oxidative stress-induced abnormalities in ceramide and cholesterol metabolism in brain aging and Alzheimer's disease. *Proc Natl Acad Sci U S A.* 2004;101: 2070-2075. doi:10.1073/pnas.0305799101
11. Rudajev V, Novotny J. Cholesterol as a key player in amyloid β -mediated toxicity in Alzheimer's disease. *Front Mol Neurosci.* 2022;15: 1-23. doi:10.3389/fnmol.2022.937056
12. Xiong H, Callaghan D, Jones A, Walker DG, Lue LF, Beach TG, et al. Cholesterol retention in Alzheimer's brain is responsible for high β - and γ -secretase activities and $A\beta$ production. *Neurobiol Dis.* 2008;29: 422-437. doi:10.1016/j.nbd.2007.10.005
13. Meng B, Abdullahi A, Ferreira IATM, Goonawardane N, Saito A, Kimura I, et al. Altered TMPRSS2 usage by SARS-CoV-2 Omicron impacts infectivity and fusogenicity. *Nature.* 2022;603: 706-714. doi:10.1038/s41586-022-04474-x
14. Kreutzberger AJB, Sanyal A, Saminathan A, Bloyet L-M, Stumpf S, Liu Z, et al. SARS-CoV-2 requires acidic pH to infect cells. *Proc Natl Acad Sci.* 2022;119: 1-12. doi:10.1073/pnas.2209514119

We thank the Reviewer for providing the list of references to further improve the manuscript, **10** of which were incorporated into the revised manuscript.

References (new references according to the valuable comments of reviewer 1 are in bold)

Axfors C, Schmitt AM, Janiaud P, Van't Hooft J, Abd-Elsalam S, Abdo EF, Abella BS, Akram J, Amaravadi RK, Angus DC et al (2021) Mortality outcomes with hydroxychloroquine and chloroquine in COVID-19 from an international collaborative meta-analysis of randomized trials. *Nat Commun* 12: 2349

Cutler RG, Kelly J, Storie K, Pedersen WA, Tammara A, Hatanpaa K, Troncoso JC, Mattson MP (2004) Involvement of oxidative stress-induced abnormalities in ceramide and cholesterol metabolism in brain aging and Alzheimer's disease. *Proc Natl Acad Sci U S A* 101: 2070-2075

Essalmani R, Andreo U, Evagelidis A, Le Devehat M, Pereira Ramos OH, Fruchart Gaillard C, Susan-Resiga D, Cohen EA, Seidah NG (2023) SKI-1/S1P Facilitates SARS-CoV-2 Spike Induced Cell-to-Cell Fusion via Activation of SREBP-2 and Metalloproteases, Whereas PCSK9 Enhances the Degradation of ACE2. *Viruses* 15

Gkouskou K, Vasilogiannakopoulou T, Andreakos E, Davanos N, Gazouli M, Sanoudou D, Eliopoulos AG (2021) COVID-19 enters the expanding network of apolipoprotein E4-related pathologies. *Redox Biol* 41: 101938

Glitscher M, Hildt E (2021) Endosomal Cholesterol in Viral Infections - A Common Denominator? *Front Physiol* 12: 750544

Hikmet F, Mear L, Edvinsson A, Micke P, Uhlen M, Lindskog C (2020) The protein expression profile of ACE2 in human tissues. *Mol Syst Biol* 16: e9610

Hoertel N, Sanchez-Rico M, Kornhuber J, Gulbins E, Reiersen AM, Lenze EJ, Fritz BA, Jalali F, Mills EJ, Cougoule C *et al* (2022) Antidepressant Use and Its Association with 28-Day Mortality in Inpatients with SARS-CoV-2: Support for the FIASMA Model against COVID-19. *J Clin Med* 11

Kang CK, Shin HM, Park WB, Kim HR (2022) Why are children less affected than adults by severe acute respiratory syndrome coronavirus 2 infection? *Cell Mol Immunol* 19: 555-557

Kreutzberger AJB, Sanyal A, Saminathan A, Bloyet LM, Stumpf S, Liu Z, Ojha R, Patjas MT, Geneid A, Scanavachi G *et al* (2022) SARS-CoV-2 requires acidic pH to infect cells. *Proc Natl Acad Sci U S A* 119: e2209514119

Kumar R, Sharma A, Srivastava JK, Siddiqui MH, Uddin MS, Aleya L (2021) Hydroxychloroquine in COVID-19: therapeutic promises, current status, and environmental implications. *Environ Sci Pollut Res Int* 28: 40431-40444

Kummer S, Lander A, Goretzko J, Kirchoff N, Rescher U, Schloer S (2022) Pharmacologically induced endolysosomal cholesterol imbalance through clinically licensed drugs itraconazole and fluoxetine impairs Ebola virus infection in vitro. *Emerg Microbes Infect* 11: 195-207

Kuo CL, Pilling LC, Atkins JL, Masoli JAH, Delgado J, Kuchel GA, Melzer D (2020) APOE e4 Genotype Predicts Severe COVID-19 in the UK Biobank Community Cohort. *J Gerontol A Biol Sci Med Sci* 75: 2231-2232

Meng B, Abdullahi A, Ferreira I, Goonawardane N, Saito A, Kimura I, Yamasoba D, Gerber PP, Fatihi S, Rathore S *et al* (2022) Altered TMPRSS2 usage by SARS-CoV-2 Omicron impacts infectivity and fusogenicity. *Nature* 603: 706-714

Navarese EP, Podhajski P, Gurbel PA, Grzelakowska K, Ruscio E, Tantry U, Magielski P, Kubica A, Niezgodna P, Adamski P *et al* (2023) PCSK9 Inhibition During the Inflammatory Stage of SARS-CoV-2 Infection. *J Am Coll Cardiol* 81: 224-234

Ortiz ME, Thurman A, Pezzulo AA, Leidinger MR, Klesney-Tait JA, Karp PH, Tan P, Wohlford-Lenane C, McCray PB, Jr., Meyerholz DK (2020) Heterogeneous expression of the SARS-Coronavirus-2 receptor ACE2 in the human respiratory tract. *EBioMedicine* 60: 102976

Parker RA, Garcia R, Ryan CS, Liu X, Shipkova P, Livanov V, Patel P, Ho SP (2013) Bile acid and sterol metabolism with combined HMG-CoA reductase and PCSK9 suppression. *J Lipid Res* 54: 2400-2409

Raut P, Waters H, Zimmerman J, Obeng B, Gosse J, Hess ST (2022) Localization-Based Super-Resolution Microscopy Reveals Relationship between SARS-CoV2 Spike and Phosphatidylinositol (4,5)-bisphosphate. *Proc SPIE Int Soc Opt Eng* 11965

Rudajev V, Novotny J (2022) Cholesterol as a key player in amyloid beta-mediated toxicity in Alzheimer's disease. *Front Mol Neurosci* 15: 937056

Stewart TG, Rebolledo PA, Mourad A, Lindsell CJ, Boulware DR, McCarthy MW, Thicklin F, Garcia Del Sol IT, Bramante CT, Lenert LA *et al* (2023) Higher-Dose Fluvoxamine and Time to Sustained Recovery in Outpatients With COVID-19: The ACTIV-6 Randomized Clinical Trial. *JAMA* 330: 2354-2363

Stiekema LCA, Willemsen L, Kaiser Y, Prange KHM, Wareham NJ, Boekholdt SM, Kuijk C, de Winther MPJ, Voermans C, Nahrendorf M *et al* (2021) Impact of cholesterol on proinflammatory monocyte production by the bone marrow. *Eur Heart J* 42: 4309-4320

Tudorache IF, Trusca VG, Gafencu AV (2017) Apolipoprotein E - A Multifunctional Protein with Implications in Various Pathologies as a Result of Its Structural Features. *Comput Struct Biotechnol J* 15: 359-365

Wang H, Yuan Z, Pavel MA, Jablonski SM, Jablonski J, Hobson R, Valente S, Reddy CB, Hansen SB (2023) The role of high cholesterol in SARS-CoV-2 infectivity. *J Biol Chem* 299: 104763

Wing PAC, Schmidt NM, Peters R, Erdmann M, Brown R, Wang H, Swadling L, Investigators CO, Newman J, Thakur N *et al* (2023) An ACAT inhibitor suppresses SARS-CoV-2 replication and boosts antiviral T cell activity. *PLoS Pathog* 19: e1011323

Xie B, Njoroge W, Dowling LM, Sule-Suso J, Cinque G, Yang Y (2022) Detection of lipid efflux from foam cell models using a label-free infrared method. *Analyst* 147: 5372-5385

Xiong H, Callaghan D, Jones A, Walker DG, Lue LF, Beach TG, Sue LI, Woulfe J, Xu H, Stanimirovic DB *et al* (2008) Cholesterol retention in Alzheimer's brain is responsible for high beta- and gamma-secretase activities and Abeta production. *Neurobiol Dis* 29: 422-437

Yuan Z, Hansen SB (2023) Cholesterol Regulation of Membrane Proteins Revealed by Two-Color Super-Resolution Imaging. *Membranes (Basel)* 13

Yuan Z, Pavel MA, Wang H, Kwachukwu JC, Mediouni S, Jablonski JA, Nettles KW, Reddy CB, Valente ST, Hansen SB (2022) Hydroxychloroquine blocks SARS-CoV-2 entry into the endocytic pathway in mammalian cell culture. *Commun Biol* 5: 958

Reviewer #2 (Comments to the Authors (Required)):

A review should aim at covering the subject matter with conciseness, fair bibliographic coverage and discussion that summarises the state-of-the-art of the field and its future directions. This reviews encompasses a rather limited analysis of the field and its bibliographic coverage is also rather limited, ignoring various recent contributions with detailed mechanistic hypothesis on the influence of cholesterol on SARS-CoV-1 cell-surface recognition, endocytosis and subsequent steps. The review is not suitable in its current state.

We acknowledge the Reviewer's critical assessment of the submitted review article. We would like to point out that the opinion of this Reviewer is in striking contrast to the editor's invitation to resubmit a revised manuscript and the assessments provided by Reviewer #1 (*'The review is well written, covers most of the key points of cholesterol as a target of COVID19 treatment, and it is timely. I recommend publication.'*) and Reviewer #3 (*'I want to thank the authors to bring this timely topic up, and I really like the overall structure of the manuscript.'*; *'The presented manuscript is well-written and covers a timely topic.'*).

As recognized by Reviewer #1, this review aimed to highlight key molecular players and mechanism at the plasma membrane and in late endosomes/lysosomes that facilitate cell entry of SARS-CoV-2 in a cholesterol-sensitive manner. The number of COVID-19-related studies addressing these and other aspects of cholesterol in COVID-19, such as the prognostic value of systemic cholesterol levels, the use of statins and other cholesterol-lowering drugs, cholesterol and viral replication, and links between SREBP1/2, inflammation and SARS-CoV2 propagation is immense. It would have been beyond the scope of this review to cover in detail all the mechanistic insights derived from these studies, often not directly related to SARS-CoV-2 cell entry, but other parts of the virus life cycle, such as replication, propagation, virus assembly and/or budding/viral release.

In the revised manuscript, 174 references are provided in the reference list, which gives readers the opportunity for further reading on each aspect of the review. From those, most references were published in the last 5 years (>2019). In addition to original research publications, this includes a substantial number of recent review articles that cover various cholesterol-related mechanisms that influence SARS-CoV2 cell entry at the surface and in late endosomes/lysosomes (e.g. Ballot et al., 2020; Cesar-Silva et al. 2022; Fecchi et al., 2020; Glitscher and Hilt, 2021; Tang et al., 2020; Wang et al., 2008, 2021, 2022 and 2023). We believe that this extensive list of recent literature provides a solid overview on the state-of-the-art of the field as well as many potential future directions related to the key aspects covered in this review article. Approximately 40 articles in the reference list cover earlier work relevant to this research field (2002-2018) and allow the reader to obtain information on earlier influential publications and how the knowledge in the field evolved over time.

As part of the Revision addressing the comments of Reviewer #1 and #3, we have added 14 mostly recent references (2020-23) to this manuscript, providing further detail and addressing additional aspects related to cholesterol and SARS-CoV-2 infectivity.

In addition, we have added a new paragraph to the Conclusions on page 15. This included the addition of 5 new references. The reference list was corrected accordingly:

‘ In this review, we described several key roles how cholesterol at the cell surface and in endolysosomes influences the efficacy of SARS-CoV-2 entry, providing opportunities for drugs targeting cholesterol-sensitive mechanism in these locations to reduce SARS-CoV-2 infectivity and COVID-19 disease severity. It would have gone beyond the scope of this review to cover the influence of cholesterol on many other aspects of the viral cycle, such as replication, assembly and viral release and we refer the reader to excellent review articles that cover cholesterol-sensitive mechanisms not only during SARS-CoV2 surface recognition and cell entry, but also viral replication, assembly and release (Ahmad *et al*, 2023; Ballout *et al*, 2020; Barrantes, 2022; Cesar-Silva *et al*, 2022; Fecchi *et al*, 2020; Glitscher & Hildt, 2021; Kowalska *et al*, 2022; Palacios-Rapalo *et al*, 2021; Tang *et al*, 2021; Wang *et al*, 2023; Wang *et al*, 2022).’

An overview of the role of Niemann-pick C1 (NPC1) in viral infections and inhibition of viral infections through NPC1 inhibitor. Ahmad I, Fatemi SN, Ghaheri M, Rezvani A, Khezri DA, Natami M, Yasamineh S, Gholizadeh O, Bahmanyar Z. *Cell Commun Signal.* 2023 Dec 14;21(1):352.

Possible mechanisms of cholesterol elevation aggravating COVID-19. Tang Y, Hu L, Liu Y, Zhou B, Qin X, Ye J, Shen M, Wu Z, Zhang P. *Int J Med Sci.* 2021 Aug 21;18(15):3533-3543.

The Influence of SARS-CoV-2 Infection on Lipid Metabolism-The Potential Use of Lipid-Lowering Agents in COVID-19 Management. Kowalska K, Sabatowska Z, Forycka J, Młynarska E, Franczyk B, Rysz J. *Biomedicines.* 2022 Sep 18;10(9):2320. d

The constellation of cholesterol-dependent processes associated with SARS-CoV-2 infection. Barrantes FJ. *Prog Lipid Res.* 2022 Jul;87:101166. doi: 10.1016/j.plipres.2022.101166.

Cholesterol-Rich Lipid Rafts as Platforms for SARS-CoV-2 Entry. Palacios-Rápalo SN, De Jesús-González LA, Cordero-Rivera CD, Farfan-Morales CN, Osuna-Ramos JF, Martínez-Mier G, Quistián-Galván J, Muñoz-Pérez A, Bernal-Dolores V, Del Ángel RM, Reyes-Ruiz JM. *Front Immunol.* 2021 Dec 16;12:796855. doi: 10.3389/fimmu.2021.796855. eCollection 2021.

Reviewer #3 (Comments to the Authors (Required)):

In the manuscript entitled "Cholesterol and COVID-19 - therapeutic opportunities at the host/virus interface during cell entry", the authors provide a summary of the current knowledge in the field of cholesterol and COVID-19 and the current achievements in drug repurposing. Targeting host factors that support virus entry, replication and/or propagation are considered as highly effective, helping to lower viral titers and to ameliorate COVID-19 severity. As many enveloped viruses, such as SARS-CoV-2, are hijacking the cellular transport machinery and fuse with the endolysosomal membrane to get access to the cell, pharmaceuticals that impede viral fusion or modulate lipid composition of the respective cellular compartments are very promising candidates for antiviral therapies.

I want to thank the authors to bring this timely topic up, and I really like the overall structure of the manuscript. I have only a few suggestions for the authors for improving the manuscript.

We thank the Reviewer for the very positive, kind and helpful comments on our review article.

1) The antiviral potential of NPC-1 inhibition and fluoxetine was recently evaluated using a more advanced human and murine 3D model (PMID: 36000328). I suggest to include the reference to bridge the gap between cell culture, animal models and the human system.

We thank the Reviewer for this suggestion and added this reference (Schloer et al. 2022) at the beginning of the 2nd para on page 10. The reference list was corrected accordingly.

2) The clinical use of the two antidepressants with high FIASMA activity has been reported to reduce SARS-CoV-2 infections and COVID-19-related mortality in inpatients, and may be appropriate for prophylaxis and/or COVID-19 therapy for outpatients or inpatients (PMID: 36233753). This would strengthen the translational aspect of the comprised research. In this context, the following reference might also be of interest (PMID: 37976072).

We thank the Reviewer for this suggestion and added a section describing these findings on page 10. The two new references were added and the reference list was corrected accordingly:

‘A retrospective cohort study of almost 400,000 hospitalized COVID-19 patients suggested that prior use of antidepressant medications may reduce the likelihood of SARS-CoV-2 infection, hospitalization and mortality. These associations between antidepressants and COVID-19 severity is most likely due to inhibition of acid sphingomyelinase (FIASMA) (Hoertel *et al.*, 2022). Yet, fluvoxamine treatment of outpatients with mild to moderate COVID-19 did not improve the time to sustained recovery (Stewart *et al.*, 2023), pointing at the need for further studies to evaluate the clinical effects of fluvoxamine in COVID-19.’

3) Interestingly, pharmacological induced endolysosomal cholesterol imbalance through the clinically licensed drugs itraconazole and fluoxetine were reported to impair another virus with pandemic potential: Ebola virus (PMID: 34919035). As Ebola virus entry requires the cholesterol transporter Niemann-Pick C1 (PMID: 26771495), the use of these cholesterol affecting drugs might be promising for future pandemic management of other enveloped viruses that use the same entry route. The authors should include a small paragraph in the conclusions to outline the broad antiviral effect of targeting cellular cholesterol levels.

The presented manuscript is well-written and covers a timely topic. I hope that the authors can provide a revised manuscript including my suggestions.

We thank the Reviewer for this suggestion and added a section describing these findings at the end of the Conclusions on page 16. The two new references were added and the reference list was corrected accordingly:

‘Cholesterol plays a crucial role during multiple steps in the life cycle of almost all viruses. Thus, manipulating cholesterol homeostasis could potentially serve as a broad-spectrum antiviral strategy. As outlined in this review, rather than reducing cholesterol levels globally, the key to cholesterol-related antiviral strategies may require disruption of cholesterol handling in specific cellular organelles where it is needed for viral entry, replication and propagation (Glitscher & Hildt, 2021).

Endolysosomal cholesterol homeostasis has been recognized as a potential target for antiviral treatments against a variety of viruses, including SARS-CoV-2, but also influenza A and Ebola viruses. In the case of Ebola, NPC1 is hijacked by the virus to cross the endolysosomal membrane for cell entry. Hence, drugs interfering with NPC1 activity and endolysosomal cholesterol levels such as itraconazole and fluoxetine have potential to deliver broad antiviral effects (Glitscher & Hildt, 2021; Kummer *et al*, 2022). Although the availability of these tools offer promising treatment options against viral infections, further research is still needed to fully understand the impact of interfering with cholesterol transport in cells and to assess adverse effects of these drugs (Glitscher & Hildt, 2021).’

Cited Literature

- Ahmad I, Fatemi SN, Ghaheri M, Rezvani A, Khezri DA, Natami M, Yasamineh S, Gholizadeh O, Bahmanyar Z (2023) An overview of the role of Niemann-pick C1 (NPC1) in viral infections and inhibition of viral infections through NPC1 inhibitor. *Cell Commun Signal* 21: 352
- Axfors C, Schmitt AM, Janiaud P, Van't Hooft J, Abd-Elsalam S, Abdo EF, Abella BS, Akram J, Amaravadi RK, Angus DC *et al* (2021) Mortality outcomes with hydroxychloroquine and chloroquine in COVID-19 from an international collaborative meta-analysis of randomized trials. *Nat Commun* 12: 2349
- Ballout RA, Sviridov D, Bukrinsky MI, Remaley AT (2020) The lysosome: A potential juncture between SARS-CoV-2 infectivity and Niemann-Pick disease type C, with therapeutic implications. *FASEB J* 34: 7253-7264
- Barrantes FJ (2022) The constellation of cholesterol-dependent processes associated with SARS-CoV-2 infection. *Prog Lipid Res* 87: 101166
- Cesar-Silva D, Pereira-Dutra FS, Moraes Giannini AL, Jacques GdAC (2022) The Endolysosomal System: The Acid Test for SARS-CoV-2. *Int J Mol Sci* 23
- Cutler RG, Kelly J, Storie K, Pedersen WA, Tammara A, Hatanpaa K, Troncoso JC, Mattson MP (2004) Involvement of oxidative stress-induced abnormalities in ceramide and cholesterol metabolism in brain aging and Alzheimer's disease. *Proc Natl Acad Sci U S A* 101: 2070-2075
- Essalmani R, Andreo U, Evagelidis A, Le Devehat M, Pereira Ramos OH, Fruchart Gaillard C, Susan-Resiga D, Cohen EA, Seidah NG (2023) SKI-1/S1P Facilitates SARS-CoV-2 Spike Induced Cell-to-Cell Fusion via Activation of SREBP-2 and Metalloproteases, Whereas PCSK9 Enhances the Degradation of ACE2. *Viruses* 15
- Fecchi K, Anticoli S, Peruzzo D, Iessi E, Gagliardi MC, Matarrese P, Ruggieri A (2020) Coronavirus Interplay With Lipid Rafts and Autophagy Unveils Promising Therapeutic Targets. *Front Microbiol* 11: 1821
- Gkouskou K, Vasilogiannakopoulou T, Andreakos E, Davanos N, Gazouli M, Sanoudou D, Eliopoulos AG (2021) COVID-19 enters the expanding network of apolipoprotein E4-related pathologies. *Redox Biol* 41: 101938
- Glitscher M, Hildt E (2021) Endosomal Cholesterol in Viral Infections - A Common Denominator? *Front Physiol* 12: 750544
- Hikmet F, Mear L, Edvinsson A, Micke P, Uhlen M, Lindskog C (2020) The protein expression profile of ACE2 in human tissues. *Mol Syst Biol* 16: e9610
- Hoertel N, Sanchez-Rico M, Kornhuber J, Gulbins E, Reiersen AM, Lenze EJ, Fritz BA, Jalali F, Mills EJ, Cougoule C *et al* (2022) Antidepressant Use and Its Association with 28-Day Mortality in Inpatients with SARS-CoV-2: Support for the FIASMA Model against COVID-19. *J Clin Med* 11
- Kang CK, Shin HM, Park WB, Kim HR (2022) Why are children less affected than adults by severe acute respiratory syndrome coronavirus 2 infection? *Cell Mol Immunol* 19: 555-557
- Kowalska K, Sabatowska Z, Forycka J, Mlynarska E, Franczyk B, Rysz J (2022) The Influence of SARS-CoV-2 Infection on Lipid Metabolism-The Potential Use of Lipid-Lowering Agents in COVID-19 Management. *Biomedicines* 10
- Kreutzberger AJB, Sanyal A, Saminathan A, Bloyet LM, Stumpf S, Liu Z, Ojha R, Patjas MT, Geneid A, Scanavachi G *et al* (2022) SARS-CoV-2 requires acidic pH to infect cells. *Proc Natl Acad Sci U S A* 119: e2209514119
- Kumar R, Sharma A, Srivastava JK, Siddiqui MH, Uddin MS, Aleya L (2021) Hydroxychloroquine in COVID-19: therapeutic promises, current status, and environmental implications. *Environ Sci Pollut Res Int* 28: 40431-40444
- Kummer S, Lander A, Goretzko J, Kirchoff N, Rescher U, Schloer S (2022) Pharmacologically induced endolysosomal cholesterol imbalance through clinically licensed drugs itraconazole and fluoxetine impairs Ebola virus infection in vitro. *Emerg Microbes Infect* 11: 195-207
- Kuo CL, Pilling LC, Atkins JL, Masoli JAH, Delgado J, Kuchel GA, Melzer D (2020) APOE e4 Genotype Predicts Severe COVID-19 in the UK Biobank Community Cohort. *J Gerontol A Biol Sci Med Sci* 75: 2231-2232

Meng B, Abdullahi A, Ferreira I, Goonawardane N, Saito A, Kimura I, Yamasoba D, Gerber PP, Fatih S, Rathore S *et al* (2022) Altered TMPRSS2 usage by SARS-CoV-2 Omicron impacts infectivity and fusogenicity. *Nature* 603: 706-714

Navarese EP, Podhajski P, Gurbel PA, Grzelakowska K, Ruscio E, Tantry U, Magielski P, Kubica A, Niezgoda P, Adamski P *et al* (2023) PCSK9 Inhibition During the Inflammatory Stage of SARS-CoV-2 Infection. *J Am Coll Cardiol* 81: 224-234

Ortiz ME, Thurman A, Pezzulo AA, Leidinger MR, Klesney-Tait JA, Karp PH, Tan P, Wohlford-Lenane C, McCray PB, Jr., Meyerholz DK (2020) Heterogeneous expression of the SARS-Coronavirus-2 receptor ACE2 in the human respiratory tract. *EBioMedicine* 60: 102976

Palacios-Rapalo SN, De Jesús-González LA, Cordero-Rivera CD, Farfan-Morales CN, Osuna-Ramos JF, Martínez-Mier G, Quistián-Galván J, Muñoz-Pérez A, Bernal-Dolores V, del Angel RM *et al* (2021) Cholesterol-Rich Lipid Rafts as Platforms for SARS-CoV-2 Entry. *Frontiers in Immunology* 12

Parker RA, Garcia R, Ryan CS, Liu X, Shipkova P, Livanov V, Patel P, Ho SP (2013) Bile acid and sterol metabolism with combined HMG-CoA reductase and PCSK9 suppression. *J Lipid Res* 54: 2400-2409

Raut P, Waters H, Zimmermanberg J, Obeng B, Gosse J, Hess ST (2022) Localization-Based Super-Resolution Microscopy Reveals Relationship between SARS-CoV2 Spike and Phosphatidylinositol (4,5)-bisphosphate. *Proc SPIE Int Soc Opt Eng* 11965

Rudajev V, Novotny J (2022) Cholesterol as a key player in amyloid beta-mediated toxicity in Alzheimer's disease. *Front Mol Neurosci* 15: 937056

Stewart TG, Rebolledo PA, Mourad A, Lindsell CJ, Boulware DR, McCarthy MW, Thicklin F, Garcia Del Sol IT, Bramante CT, Lenert LA *et al* (2023) Higher-Dose Fluvoxamine and Time to Sustained Recovery in Outpatients With COVID-19: The ACTIV-6 Randomized Clinical Trial. *JAMA* 330: 2354-2363

Stiekema LCA, Willemsen L, Kaiser Y, Prange KHM, Wareham NJ, Boekholdt SM, Kuijk C, de Winther MPJ, Voermans C, Nahrendorf M *et al* (2021) Impact of cholesterol on proinflammatory monocyte production by the bone marrow. *Eur Heart J* 42: 4309-4320

Tang Y, Hu L, Liu Y, Zhou B, Qin X, Ye J, Shen M, Wu Z, Zhang P (2021) Possible mechanisms of cholesterol elevation aggravating COVID-19. *Int J Med Sci* 18: 3533-3543

Tudorache IF, Trusca VG, Gafencu AV (2017) Apolipoprotein E - A Multifunctional Protein with Implications in Various Pathologies as a Result of Its Structural Features. *Comput Struct Biotechnol J* 15: 359-365

Wang H, Yuan Z, Pavel MA, Jablonski SM, Jablonski J, Hobson R, Valente S, Reddy CB, Hansen SB (2023) The role of high cholesterol in SARS-CoV-2 infectivity. *J Biol Chem* 299: 104763

Wang T, Cao Y, Zhang H, Wang Z, Man CH, Yang Y, Chen L, Xu S, Yan X, Zheng Q *et al* (2022) COVID-19 metabolism: Mechanisms and therapeutic targets. *MedComm* (2020) 3: e157

Wing PAC, Schmidt NM, Peters R, Erdmann M, Brown R, Wang H, Swadling L, Investigators CO, Newman J, Thakur N *et al* (2023) An ACAT inhibitor suppresses SARS-CoV-2 replication and boosts antiviral T cell activity. *PLoS Pathog* 19: e1011323

Xie B, Njoroge W, Dowling LM, Sule-Suso J, Cinque G, Yang Y (2022) Detection of lipid efflux from foam cell models using a label-free infrared method. *Analyst* 147: 5372-5385

Xiong H, Callaghan D, Jones A, Walker DG, Lue LF, Beach TG, Sue LI, Woulfe J, Xu H, Stanimirovic DB *et al* (2008) Cholesterol retention in Alzheimer's brain is responsible for high beta- and gamma-secretase activities and A β production. *Neurobiol Dis* 29: 422-437

Yuan Z, Hansen SB (2023) Cholesterol Regulation of Membrane Proteins Revealed by Two-Color Super-Resolution Imaging. *Membranes (Basel)* 13

Yuan Z, Pavel MA, Wang H, Kwachukwu JC, Mediouni S, Jablonski JA, Nettles KW, Reddy CB, Valente ST, Hansen SB (2022) Hydroxychloroquine blocks SARS-CoV-2 entry into the endocytic pathway in mammalian cell culture. *Commun Biol* 5: 958

February 5, 2024

RE: Life Science Alliance Manuscript #LSA-2023-02453-TR

Prof. Christa Buechler
University Hospital Regensburg
Franz Josef Strauss Allee 11
Regensburg 93042
Germany

Dear Dr. Buechler,

Thank you for submitting your revised manuscript entitled "Cholesterol and COVID-19 - therapeutic opportunities at the host/virus interface during cell entry". We would be happy to publish your paper in Life Science Alliance pending final revisions necessary to meet our formatting guidelines.

- please be sure that the authorship listing and order is correct
- please add the Twitter handle of your host institute/organization as well as your own or/and one of the authors in our system
- please upload your manuscript file without track changes
- please remove figures from the manuscript file
- please add your figure legends to the main manuscript text after the references section

A. FINAL FILES:

B. MANUSCRIPT ORGANIZATION AND FORMATTING:

Sincerely,

Reviewer #1 (Comments to the Authors (Required)):

The authors have more than adequately addressed all my concerns. It is a timely well written review and I recommend publication.

February 13, 2024

RE: Life Science Alliance Manuscript #LSA-2023-02453-TRR

Prof. Christa Buechler
University Hospital Regensburg
Franz Josef Strauss Allee 11
Regensburg 93042
Germany

Dear Dr. Buechler,

Thank you for submitting your Review entitled "Cholesterol and COVID-19 - therapeutic opportunities at the host/virus interface during cell entry". It is a pleasure to let you know that your manuscript is now accepted for publication in Life Science Alliance. Congratulations on this interesting work.

Again, congratulations on a very nice paper. I hope you found the review process to be constructive and are pleased with how the manuscript was handled editorially. We look forward to future exciting submissions from your lab.

Sincerely,
